# ANALYSIS OF A CLASS OF STOCHASTIC COMPONENT-WISE SOFT-CLIPPING SCHEMES

## ABSTRACT

Choosing the optimization algorithm that performs best on a given machine learning problem is often delicate, and there is no guarantee that current state-of-the-art algorithms will perform well across all tasks. Consequently, the more reliable methods that one has at hand, the larger the likelihood of a good end result. To this end, we introduce and analyze a large class of stochastic so-called soft clipping schemes with a broad range of applications. Despite the wide adoption of clipping techniques in practice, soft clipping methods have not been analyzed to a large extent in the literature. In particular, a rigorous mathematical analysis is lacking in the general, nonlinear case. Our analysis lays a theoretical foundation for a large class of such schemes, and motivates their usage.

In particular, under standard assumptions such as Lipschitz continuous gradients of the objective function, we give rigorous proofs of convergence in expectation with rates in both the convex and the non-convex case, as well as almost sure convergence to a stationary point in the non-convex case. The computational cost of the analyzed schemes is essentially the same as that of stochastic gradient descent.

## 1 INTRODUCTION

In this article we consider the problem

$$w_* = \arg\min_{w \in \mathbb{R}^d} F(w), \tag{1}$$

where $F \colon \mathbb{R}^d \to \mathbb{R}$ is an objective function of the form

$$F(w) = \mathbb{E}\left[f(w, \xi)\right], \tag{2}$$

with a random variable $\xi$. A common setting in machine learning is that the objective function is given by

$$F(w) = \frac{1}{N} \sum_{i=1}^{N} \ell(m(x_i, w), y_i). \tag{3}$$

Here $\{(x_i, y_i)\}_{i=1}^{N} \in \mathcal{X} \times \mathcal{Y}$ is a data set with features $x_i$ in a feature space $\mathcal{X}$ and labels $y_i$ in a label space $\mathcal{Y}$, $\ell$ is a loss function and $m(\cdot, w)$ is a model (such as a neural network) with parameters $w$. In this case, $f(w, \xi)$ typically corresponds to evaluating randomly chosen parts of the sum.

A widely adopted method for solving problems of the type (2), when the objective function is given by (3), is the *stochastic gradient descent* (SGD) algorithm

$$w_{k+1} = w_k - \alpha_k \nabla f(w_k, \xi_k),$$

first introduced in the seminal work Robbins & Monro (1951). Despite its many advantages, such as being less computationally expensive than the usual gradient descent algorithm, and its ability to escape local saddle points (Fang et al., 2019), two well known shortcomings of SGD is its sensitivity to the choice of step size/learning rate $\alpha_k$ and its inaptitude for *stiff* problems (Owens & Filkin, 1989). For instance, Andradóttir (1990) showed that the iterates may grow explosively, if the function suffers from steep gradients and if the initial step size is not chosen properly. A simple

example is when the objective function to be minimized is given by $F(w) = \frac{w^4}{4}, w \in \mathbb{R}$. It can be shown by induction that if the initial iterate $w_1 \geq \sqrt{3/\alpha_0}$ and $\alpha_k = \alpha_0/k$, the iterates will satisfy $|w_k| \geq |w_1|k!$, compare Andradóttir (1990, Lemma 1). Furthermore, as noted in Nemirovski et al. (2009), even in the benign case when the objective function is strongly convex and convergence is guaranteed, the convergence can be extremely slow with an ill-chosen step size. Moreover, it has long been known that the loss landscape of neural networks can have steep regions where the gradients become very large, so-called *exploding gradients* (Goodfellow et al., 2016; Pascanu et al., 2013). Hence, the complications illustrated by the previous examples are not merely theoretical, but constitute practical challenges that one encounters when training neural networks.

The concept of *gradient clipping*, i.e. rescaling the gradient increments, is often used to alleviate the issue of steep gradients. For stiff problems, where different components of the solution typically evolve at different speeds, we also expect that rescaling different components in different ways will improve the behaviour of the method. In this paper, we therefore combine these concepts and consider a general class of soft-clipped, componentwise stochastic optimization schemes. More precisely, we consider the update

$$
\begin{aligned}
w_{k+1} &= w_k - \alpha_k G\big(\nabla f(w_k, \xi_k), \alpha_k\big) \\
&= w_k - \alpha_k \nabla f(w_k, \xi_k) + \alpha_k^2 H(\nabla f(w_k, \xi_k), \alpha_k),
\end{aligned}
\tag{4}
$$

where $G$ and $H$ are operators that apply functions $g, h \colon \mathbb{R} \times \mathbb{R} \to \mathbb{R}$ component-wise to the gradient $\nabla f(w_k, \xi_k) \in \mathbb{R}^d$. Essentially the functions $g$ and $h$ are generalizations of the clipping functions stated in (8) and (9) below, respectively. We note that either of these functions completely specifies the scheme; it is natural to specify $G$, and once this is done $H$ can be determined by algebraic manipulations. We further note that if $\alpha_k$ is allowed to be a vector, one might equivalently see the clipping as a re-scaling of the components $(\alpha_k)_i$ rather than of the gradient components. However, here we choose to rescale the gradient, and our $\alpha_k$ is therefore a scalar.

## 2 RELATED WORKS

The concepts of *gradient clipping* and the related *gradient normalization* are not new. In the case of gradient normalization, the increment is re-scaled to have unit norm:

$$
w_{k+1} = w_k - \alpha_k \frac{\nabla F(w_k)}{\|\nabla F(w_k)\|}.
$$

This scheme is mentioned early in the optimization literature, see Poljak (1967), and a stochastic counterpart appears already in Azadivar & Talavage (1980). A version of the latter, in which two independent approximations to the gradient are sampled and then normalized, was proposed and analyzed in Andradóttir (1990; 1996).

A related idea is that of *gradient clipping*, see e.g Goodfellow et al. (2016, Sec. 10.11.1.). In the context of neural networks, this idea was proposed in Pascanu et al. (2013). In so-called hard clipping (Zhang et al., 2020a), the gradient approximation is simply projected onto a ball of predetermined size $\gamma_k$;

$$
w_{k+1} = w_k - \alpha_k \min\Big(1, \frac{\gamma_k}{\|\nabla f(w_k, \xi_k)\|}\Big) \nabla f(w_k, \xi_k).
\tag{5}
$$

A momentum version of this scheme for convex functions with quadratic growth as well as weakly convex functions, was analyzed in Mai & Johansson (2021). A similar algorithm was proposed and analyzed in Zhang et al. (2020a) under a relaxed differentiability condition on the gradient – the $(L_0, L_1)$ – smoothness condition, which was introduced in Zhang et al. (2020b). Yet another example is Gorbunov et al. (2020), who derives high-probability complexity bounds for an accelerated, clipped version of SGD with heavy-tailed distributed noise in the stochastic gradients.

A drawback of the rescaling in (5) is that it is not a differentiable function of $\|\nabla f(w_k, \xi_k)\|$. A smoothed version where this is the case is instead given by

$$
\begin{aligned}
w_{k+1} &= w_k - \frac{\alpha_k}{1 + \alpha_k \|\nabla f(w_k, \xi_k)\|/\gamma_k} \nabla f(w_k, \xi_k) \\
&= w_k - \frac{\alpha_k \gamma_k}{\gamma_k + \alpha_k \|\nabla f(w_k, \xi_k)\|} \nabla f(w_k, \xi_k)
\end{aligned}
\tag{6}
$$

and referred to as soft clipping, see Zhang et al. (2020a). It was observed in Zhang et al. (2020a) that soft clipping results in a smoother loss curve, which indicates that the learning process is more robust and less sensitive to noise in the underlying data set. This makes soft clipping algorithms a more desirable alternative than hard clipping. How to choose $\gamma_k$ is, however, not clear a priori, and some convergence analyses require that it grows as $\alpha_k$ decreases. With the choice $\gamma_k \equiv 1$, we acquire the tamed SGD method independently introduced in Eisenmann & Stillfjord (2022). This method is based on the tamed Euler scheme for approximating solutions to stochastic differential equations, and given by

$$w_{k+1} = w_k - \frac{\alpha_k \nabla f(w_k, \xi_k)}{1 + \alpha_k \|\nabla f(w_k, \xi_k)\|}.$$

We note that (6) can be equivalently stated as

$$w_{k+1} = w_k - \alpha_k \nabla f(w_k, \xi_k) + \alpha_k^2 \frac{\|\nabla f(w_k, \xi_k)\| \nabla f(w_k, \xi_k)}{\gamma_k + \alpha_k \|\nabla f(w_k, \xi_k)\|}, \tag{7}$$

i.e. it is a second-order perturbation of SGD, compare Eisenmann & Stillfjord (2022). As $\alpha_k \to 0$, the method thus behaves more and more like SGD. This is a desirable feature due to the many good properties that SGD has as long as it is stable.

The problem in (1) can be restated as finding stationary points of the gradient flow equation

$$\dot{w}(t) = -\nabla F(w(t)), \ t \in \mathbb{R}_+.$$

An issue that one frequently encounters when solving such ordinary differential equations numerically is that of *stiffness*, see Söderlind et al. (2015). In the case when $F$ is given by a neural network, this translates to the fact that different components of the parameters converge at different speed and have different step size restrictions, see Owens & Filkin (1989). An approach sometimes used in stochastic optimization algorithms that mitigates this issue, is that of performing the gradient update element-wise, see for example Mikolov (2013); Duchi et al. (2011); Kingma & Ba (2015). With $(w_k)_i$ denoting the $i$th component of the vector $w_k$, an element-wise version of update (6) could for example be stated as

$$(w_{k+1})_i = (w_k)_i - \frac{\alpha_k \gamma_k \frac{\partial f(w_k, \xi_k)}{\partial x_i}}{\gamma_k + \alpha_k \left| \frac{\partial f(w_k, \xi_k)}{\partial x_i} \right|}. \tag{8}$$

Using the reformulation (7), this can be further rewritten as

$$(w_{k+1})_i = (w_k)_i - \alpha_k \frac{\partial f(w_k, \xi_k)}{\partial x_i} + \alpha_k^2 h \left( \frac{\partial f(w_k, \xi_k)}{\partial x_i}, \alpha_k \right),$$

where

$$h(x_i, \alpha_k) = \frac{|x_i| x_i}{\gamma_k + \alpha_k |x_i|}. \tag{9}$$

## 3 CONTRIBUTIONS

Our algorithms and analysis bear similarities to those analyzed in Zhang et al. (2020a) and Mai & Johansson (2021), but while they consider standard hard clipping for momentum algorithms in their analysis, we consider general, soft-clipped algorithms versions of SGD. Under similar assumptions, they obtain convergence guarantees in the convex- and the non-convex case. The class of schemes considered here, is also reminiscent of other componentwise algorithms such as those introduced in e.g. Duchi et al. (2011); Zeiler (2012); Kingma & Ba (2015); Hinton (2018). While their emphasis is on an average regret analysis in the convex case, the focus of the analysis in this paper is on minimizing an objective function with a particular focus on the non-convex case. The algorithms in Duchi et al. (2011); Zeiler (2012); Kingma & Ba (2015); Hinton (2018) are also rather formulated as an adaptation of the step size based on information of the local cost landscape, obtained from gradient information calculated in past iterations. In contrast, our methods seeks to control the step size based on gradient data from the current iterate.

In Appendix E of Zhang et al. (2020a) it is claimed that "soft clipping is in fact equivalent to hard clipping up to a constant factor of 2" (Zhang et al., 2020a, Appendix E, p. 27). A similar claim is made in Zhang et al. (2020b, p. 5). However, it is not stated in what sense the algorithms are equivalent; in general it is not possible to rewrite a hard clipping scheme as a soft clipping scheme. The argument given in Zhang et al. (2020a) is that one can bound the norm of the gradient

$$\frac{1}{2}\min(\alpha\|\nabla f(w_k,\xi_k)\|,\gamma) \leq \alpha\frac{\|\nabla f(w_k,\xi_k)\|}{1+\alpha\|\nabla f(w_k,\xi_k)\|/\gamma} \leq \min(\alpha\|\nabla f(w_k,\xi_k)\|,\gamma)$$

and therefore the schemes are equivalent in some sense. However, the fact that the norms of the gradient are bounded or even equal at some stage does not imply that one scheme converges if the other does. As a counterexample, take $w_0 = \tilde{w}_0$ and consider $w_{k+1} = w_k - \alpha\nabla F(w_k)$ and $\tilde{w}_{k+1} = \tilde{w}_k + \alpha\nabla F(\tilde{w}_k)$. Then $\|\nabla F(w_0)\| = \|\nabla F(\tilde{w}_0)\|$, but one of them can converge while the other diverges.

In the strongly convex, case we prove that with a decreasing step size, $\mathbb{E}[F(w_k)]$ converges to the minimal function value at a rate of $\mathcal{O}(\frac{1}{K})$, where $K$ is the total number of iterations. In the non-convex case, we show that $\min_{1\leq k\leq K}\|\nabla F(w_k)\|^2$ converges to 0, in expectation, as well as almost surely. With a decreasing step size of $\alpha_k = \frac{\beta}{k+\gamma}$ this convergence is at a rate of $\mathcal{O}(\frac{1}{\log(K)})$. The main focus in this article is the decreasing step size regime, but a slight extension yields $\mathcal{O}(\frac{1}{\sqrt{K}})$−convergence when a constant step size of $\frac{1}{\sqrt{K}}$ (depending on the total number of iterations $K$) is used in the non-convex case. Similary, we obtain $\mathcal{O}(\frac{1}{K})$−convergence in the strongly convex case for a fixed step size of $\alpha_k = \frac{1}{K}$.

The analysis provides a theoretical justification for using a large class of soft clipping algorithms and our numerical experiments give further insight into their behavior and performance in general.

We investigate the behavior of the algorithms on common large scale machine learning problems, and see that their performance is essentially on par with state of the art algorithms such as Adam (Kingma & Ba, 2015) and SGD with momentum (Qian, 1999).

## 4 SETTING

Here we briefly discuss the setting that we consider for approximating a solution to (1) with the sequence $(w_k)_{k\in\mathbb{N}}$ generated by the method (4). The formal details can be found in Appendix A, since most of the assumptions that we make are fairly standard. To begin with, we assume that the sequence $\{\xi_k\}_{k\geq 1}$ in (4) is a sequence of independent, identically distributed random variables. We will frequently make use of the notation $\mathbb{E}_{\xi_k}[X]$ for the conditional expectation of $X$ with respect to all the variables $\xi_1,\ldots,\xi_{k-1}$.

For the clipping functions $G$ and $H$ in Algorithm (4), we assume that they are bounded in norm as follows; $\|G(x,\alpha)\| \leq c_g\|x\|$ and $\|H(x,\alpha)\| \leq c_h\|x\|^2$, for some constants $c_g$ and $c_h$. These assumptions are very general and allows for analyzing a large class of both component-wise and non-componentwise schemes. This is summarized in Assumption 1, with examples given in Appendix B.

Further, we assume that the stochastic gradients are unbiased estimates of the full gradient of the objective function, and that the gradient of the objective function is Lipschitz continuous. These are two very common assumptions to make in the analysis of stochastic optimization algorithms and are stated in their entirety in Assumption 2 and 3 respectively.

Similar to Eisenmann & Stillfjord (2022), we also make the reasonable assumption that there exists $w_* \in \arg\min_{w\in\mathbb{R}}$ at which the second moment is bounded, i.e.

$$\mathbb{E}\left[\|\nabla f(w_*,\xi)\|^2\right] \leq \sigma^2.$$

This is Assumption 4. As an alternative, slightly stricter assumption which improves the error bounds, we also consider an interpolation assumption, stated in Assumption 5. Essentially this says that if $w_*$ is a minimum of the objective funcion, it is also a minimum of all the stochastic functions. This is a sensible assumption for many machine learning problems, and we discuss the details in Appendix A.

For the sequence $(w_k)_{k \in \mathbb{N}}$ of increments given by the method (12), we make an additional stability assumption, specified in Assumption 6. In essence it is saying that there is a constant $M > 0$ such that the $w_* \in \arg\min_{w \in \mathbb{R}^d} F(w)$ from Assumption 4 also satisfies

$$\mathbb{E}\big[\|w_k - w_*\|^3\big] \leq M$$

for every $k \in \mathbb{N}$. Lemma 3 then shows that the previous bound holds for the exponents $q = 1, 2$ too.

## 5 CONVERGENCE ANALYSIS

In this section, we will state several convergence results for all methods that fit into the setting that is described in the previous section. A crucial first step in the proofs of our main convergence theorems are the following two lemmas. They both provide similar bounds on the per-step decrease of $\mathbb{E}\big[F(w_k)\big]$, but their sharpness differs depending on whether Assumption 4 or Assumption 5 is used.

**Lemma 1.** *Let Assumptions 1, 2, 3, 4 and 6 be fulfilled. Further, let $\{w_k\}_{k \in \mathbb{N}}$ be the sequence generated by the method (12). Then it holds that*

$$\mathbb{E}\big[F(w_{k+1})\big] - \mathbb{E}\big[F(w_k)\big] \leq -\alpha_k \mathbb{E}\big[\|\nabla F(w_k)\|^2\big] + \alpha_k^2 B_1,$$

*where*

$$B_1 = 2c_h L^3 M + 2c_h L M^{1/3} \sigma^2 + c_g^2 L^3 M^{2/3} + c_g^2 L \sigma^2.$$

Under the alternative assumption that $\nabla f(w_*, \xi_k) = 0$ a.s., we can improve the error constant as can be seen in the following lemma.

**Lemma 2.** *Let Assumptions 1, 2, 3, 5 and 6 be fulfilled. Further, let $\{w_k\}_{k \in \mathbb{N}}$ be the sequence generated by the method (12). Then it holds that*

$$\mathbb{E}\big[F(w_{k+1})\big] - \mathbb{E}\big[F(w_k)\big] \leq -\alpha_k \mathbb{E}\big[\|\nabla F(w_k)\|^2\big] + \alpha_k^2 B_2,$$

*where*

$$B_2 = c_h L^3 M + \frac{c_g^2 L^3}{2} M^{2/3}.$$

Detailed proofs of these lemmas as well as all following results are provided in Appendix C. As shown there, applying some algebraic manipulations to these auxiliary bounds and summing up quickly leads to our first main convergence result:

**Theorem 1.** *Let Assumptions 1, 2, 3 and 6 as well as Assumption 4 or Assumption 5 be fulfilled. Further, let $\{w_k\}_{k \in \mathbb{N}}$ be the sequence generated by the method (12). Then it follows that*

$$\min_{1 \leq k \leq K} \mathbb{E}\big[\|\nabla F(w_k)\|^2\big] \leq \frac{F(w_1) - F(w_*)}{\sum_{k=1}^{K} \alpha_k} + B_i \frac{\sum_{k=1}^{K} \alpha_k^2}{\sum_{k=1}^{K} \alpha_k}, \tag{10}$$

*where the constant $B_i$, $i \in \{1, 2\}$, is stated in Lemma 1 (with Assumption 4) and Lemma 2 (with Assumption 5), respectively.*

*Under the standard assumption that $\sum_{k=1}^{\infty} \alpha_k^2 < \infty$ and $\sum_{k=1}^{\infty} \alpha_k = \infty$, in particular, it holds that*

$$\lim_{K \to \infty} \min_{1 \leq k \leq K} \mathbb{E}\big[\|\nabla F(w_k)\|^2\big] = 0.$$

A standard example for a step size sequence $\{\alpha_k\}_{k \in \mathbb{N}}$ that is square summable but not summable is given by choosing $\alpha_k = \frac{\beta}{k + \gamma}$ for $\beta, \gamma \in \mathbb{R}^+$. We require these conditions to control that the step size tends to zero fast enough to compensate for the inexact gradient but slow enough such that previously made errors can still be negated by the coming steps. For this concrete example of a step size sequence, we can say more about the speed of the convergence. The parameters $\beta$ and $\gamma$ can be chosen to optimize the speed of convergence in practice. In particular, we have the following corollary:

**Corollary 1.** *Let the conditions of Theorem 1 be fulfilled and suppose that the step size sequence in (12) is defined by $\alpha_k = \frac{\beta}{k+\gamma}$. Then it follows that*

$$\min_{1 \leq k \leq K} \mathbb{E}\left[\|\nabla F(w_k)\|^2\right] \leq \frac{F(w_1) - F(w_*)}{\beta \ln(K + \gamma + 1)} + B_i \frac{\beta(2 + \gamma)}{\ln(K + \gamma + 1)},$$

*where the constant $B_i$, $i \in \{1, 2\}$, is stated in Lemma 1 (with Assumption 4) and Lemma 2 (with Assumption 5), respectively.*

Another way to obtain a converce rate, is to employ a fixed step size, but letting it depend on the total number of iterations. We get the following Corollary to Theorem 1:

**Corollary 2.** *By taking a constant step size of $\alpha_k = \frac{1}{\sqrt{K}}$, $k = 1, \ldots, K$, where $K$ is the total number of iterations, it follows that $\min_{1 \leq k \leq K} \mathbb{E}\left[\|\nabla F(w_k)\|^2\right]$ is $\mathcal{O}(\frac{1}{\sqrt{K}})$.*

Moreover, making use of Theorem 1 and the fact that the sequence in the expectation on the left-hand side of (10) is decreasing, we can conclude that it converges almost surely. This means that the probability of picking a path (or choosing a random seed) for which it does not converge is 0.

**Corollary 3.** *Let the conditions of Theorem 1 be fulfilled. It follows that the sequence*

$$\{\zeta_K(\omega)\}_{K \in \mathbb{N}}, \text{ where } \zeta_K(\omega) = \min_{1 \leq k \leq K} \|\nabla F(w_k(\omega))\|_2^2, \tag{11}$$

*converges to 0 for almost all $\omega \in \Omega$, i.e.*

$$\mathbb{P}\left(\left\{\omega \in \Omega : \lim_{K \to \infty} \zeta_K(\omega) = 0\right\}\right) = 1.$$

Our focus in this article has been on the non-convex case. This is reflected in the above convergence results which show that $\nabla F(w_k)$ tends to zero, which is essentially the best kind of convergence which can be considered. If the objective function is in addition strongly convex, there is a unique global minimum $w_*$, and it becomes possible to improve on both the kind of convergence and its speed. For example, we have the following theorem.

**Theorem 2.** *Let Assumptions 1, 2, 3 and 6 as well as Assumption 4 or Assumption 5 be fulfilled. Additionally, let F be strongly convex with convexity constant $c \in \mathbb{R}^+$, i.e.*

$$\langle \nabla F(v) - \nabla F(w), v - w \rangle \geq c\|v - w\|^2$$

*is fulfilled for all $v, w \in \mathbb{R}^d$. Further, let $\alpha_k = \frac{\beta}{k+\gamma}$ with $\beta, \gamma \in \mathbb{R}^+$ such that $\beta \in (\frac{1}{2c}, \frac{1+\gamma}{2c})$. Finally, let $\{w_k\}_{k \in \mathbb{N}}$ be the sequence generated by the method (12). Then it follows that*

$$F(w_k) - F(w_*) \leq \frac{(1 + \gamma)^{2\beta c}}{(k + 1 + \gamma)^{2\beta c}}\left(F(w_1) - F(w_*)\right) + \frac{B_i e^{\frac{2\beta c}{1+\gamma}}}{2\beta c - 1} \cdot \frac{1}{k + 1 + \gamma},$$

*where the constant $B_i$, $i \in \{1, 2\}$, is stated in Lemma 1 (with Assumption 4) and Lemma 2 (with Assumption 5), respectively.*

The proof is based on the inequality $2c(F(w_k) - F(w_*)) \leq \|\nabla F(w_k)\|^2$, see e.g. Inequality (4.12) in Bottou et al. (2018), which allows us to make the bounds in Lemma 1 and Lemma 2 explicit. See Appendix C for details.

**Remark 1.** *The constant $\gamma \in \mathbb{R}^+$ is required to get the optimal convergence rate $\mathcal{O}\left(\frac{1}{k}\right)$. With $\gamma = 0$ and $\beta = \frac{1}{2c}$ we would instead get the rate $\mathcal{O}\left(\frac{\ln k}{k}\right)$, and $\beta < \frac{1}{2c}$ results in $\mathcal{O}\left(\frac{1}{k^{2\beta c}}\right)$, cf. Theorem 5.3 in Eisenmann & Stillfjord (2022).*

**Remark 2.** *Similarly to Corollary 2 we can use Lemma 1 or Lemma 2 along with the strong convexity, to obtain that $\mathbb{E}\left[F(w_k) - F_*\right]$ is $\mathcal{O}(\frac{1}{K})$ when a constant step size of $\alpha_k = \frac{1}{K}$ is being used, compare e.g. Theorem 4.6 in Bottou et al. (2018).*

## 6 NUMERICAL EXPERIMENTS

In order to illustrate the behaviour of the different kinds of re-scaling, we set up three numerical experiments. For the implementation, we use TensorFlow (Abadi et al., 2015), version 2.12.0. The

re-scalings that we investigate corresponds to the functions in Remark 3 (with $\gamma = \frac{1}{3}$), see also Appendix B. The behavior of the methods are then studied along side those of Adam (Kingma & Ba, 2015) SGD with momentum (Qian, 1999) and component-wise, clipped SGD as implemented in Abadi et al. (2015). For Adam we use the standard parameters $\beta_1 = 0.9$ and $\beta_2 = 0.999$. SGD with momentum is run with the typical choice of a momentum of 0.9. For clipped SGD, we use a clipping factor of 1.

### 6.1 QUADRATIC COST FUNCTIONAL

In the first experiment, we consider a quadratic cost functional $F(w) = w^T A w + b^T w + c$, where $A \in \mathbb{R}^{50 \times 50}$, $b \in \mathbb{R}^{50}$ and $c = 13$. The matrix $A$ is diagonal with smallest eigenvalue $7.9 \cdot 10^{-2}$ and largest $3.8 \cdot 10^4$. Since the ratio between these is $4.9 \cdot 10^5$, this is a stiff problem. Both $A$ and $b$ are constructed as sums of other matrices and vectors, and the stochastic approximation $\nabla f(w, \xi)$ to the gradient $\nabla F(w)$ is given by taking randomly selected partial sums. The details of the setup can be found in Appendix D.

We apply the (non-componentwise) soft clipping scheme (6) and its componentwise version (8) and run each for 15 epochs (480 iterations) with different fixed step sizes $\alpha_k \equiv \alpha$. In Figure 1, we plot the final errors $\|w_{480} - z\|$ where $z = -\frac{1}{2}A^{-1}b$ is the exact solution to the minimization problem. For small step sizes, none of the rescalings do something significant and both methods essentially coincide with standard SGD. But for larger step sizes, we observe that the componentwise version outperforms the standard soft clipping. We note that the errors are rather big because we have only run the methods for a fixed number of steps, and because the problem is challenging for any gradient-based method.

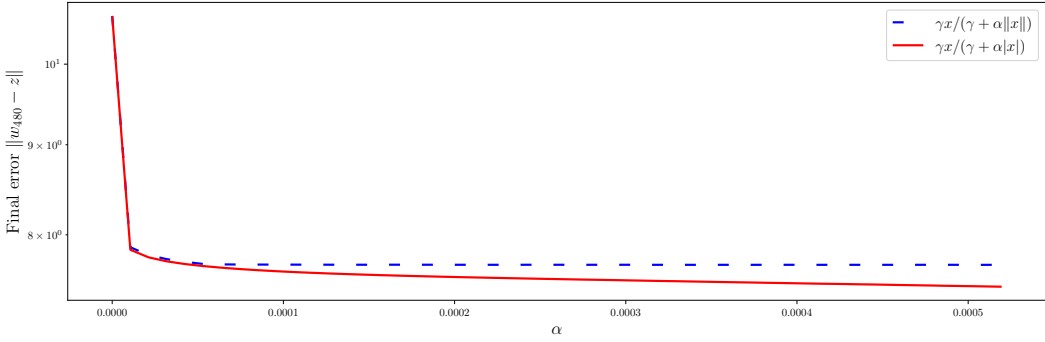

Figure 1: Final errors $\|w_{480} - z\|$ after applying componentwise (8) and non-componentwise soft clipping (6) to a stiff quadratic minimization problem. The methods are both run with a fixed step size $\alpha$ for 15 epochs, and the resulting errors for different $\alpha$ are plotted.

Additionally, we apply the different rescalings mentioned above, along with Adam, SGD, SGD with momentum and hard-clipped SGD. The results are plotted in Figure 2. We observe several things. First, we see how both SGD and SGD with momentum diverge unless the step size is very small. For SGD, the limit is given in terms of the largest eigenvalue; $2/(3.8 \cdot 10^4) \approx 5.3 \cdot 10^{-5}$, but for more complex problems it is difficult to determine a priori. Secondly, neither Adam nor the hard-clipped SGD work well for the chosen step sizes. This is notable, because 0.0001 and 0.001 are standard choices for Adam, but in this problem an initial step size of about 0.01 is required to get reasonable results. Finally, the trigonometric rescaling does not perform well, but like the other clipping schemes it does not explode. Overall, as expected, the clipping schemes are more robust to the choice of step size than the non-clipping schemes.

### 6.2 VGG NETWORK WITH CIFAR-10

In the second experiment, we construct a standard machine learning optimization problem by applying a VGG-network to the CIFAR10-data set. Details on the network and dataset are provided in

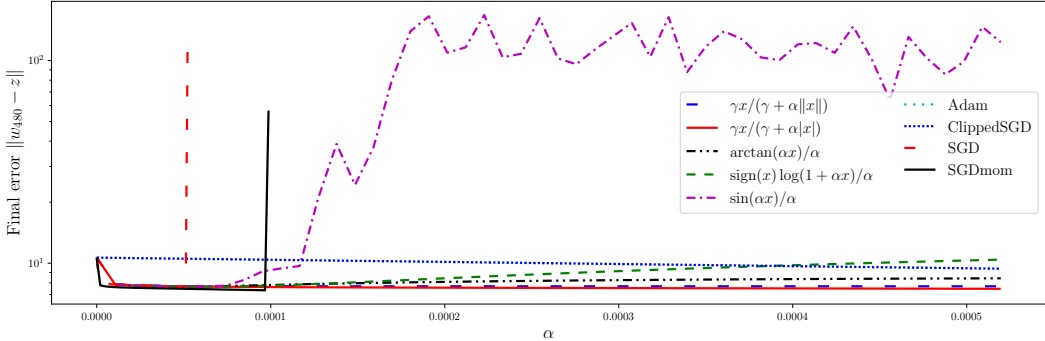

Figure 2: Final errors $\|w_{480} - z\|$ after applying different componentwise soft clipping schemes and some state-of-the-art methods to a stiff quadratic minimization problem. The methods are both run with a fixed step size $\alpha$ for 15 epochs, and the resulting errors for different $\alpha$ are plotted.

Appendix E. We use a decreasing learning rate

$$\alpha_k = \frac{\beta}{1 + 10^{-4}k},$$

where $\beta > 0$ is the initial step-size and $k$ is the iteration count (not the epoch count). After rescaling with the factor $10^4$, this has the same form as the step size in Corollary 1.

The network was trained for 150 epochs with each method and we trained it using 5 random seeds ranging from 0 to 4, similar to Zhang et al. (2020a). After this the mean of the losses and accuracies were computed. Further, we trained the network for a grid of initial step sizes with values

$$\left(10^{-5}, 5 \cdot 10^{-5}, 10^{-4}, 5 \cdot 10^{-4}, 10^{-3}, 5 \cdot 10^{-3}, 10^{-2}, 5 \cdot 10^{-2}, 0.1, 0.5, 1\right).$$

For each method, the step size with highest test accuracy was selected. In Figure 3, we see the accuracy, averaged over the random seeds. In this experiment all the algorithms give very similar results.

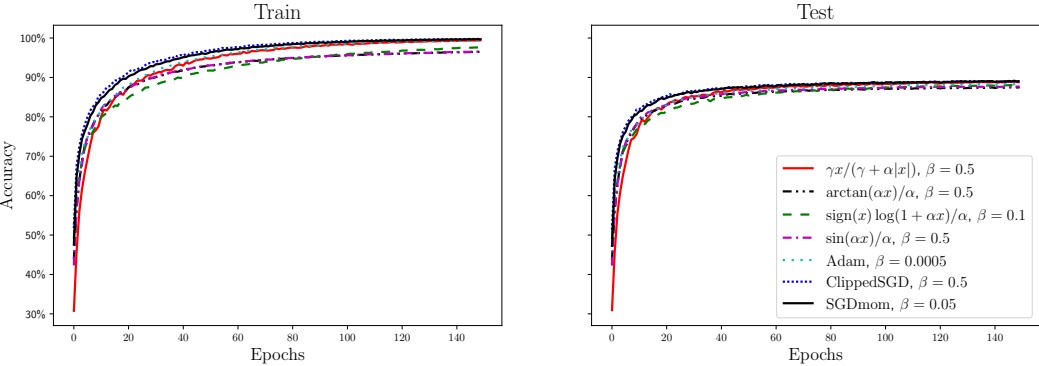

Figure 3: Accuracy of the different methods when used for training a VGG network to classify the CIFAR-10 data set. The functions in the legend correspond to the component-wise clipping functions listed in Remark 3 and in Examples 1–4 in Appendix B. By 'Adam', 'SGDmom' and 'ClippedSGD' we denote the methods Adam, SGD with momentum and component-wise, clipped SGD respectively. For all the methods, $\beta > 0$ is the initial step size parameter.

### 6.3 RNN FOR CHARACTER PREDICTION

The third experiment is a Recurrent Neural Network architecture for character-level text prediction of the Pennsylvania Treebank portion of the Wall Street Journal corpus (Marcus et al., 1993), similar

to e.g. Graves (2013); Mikolov et al. (2012); Pascanu et al. (2012). Details on the network, dataset and the so-called perplexity measure of accuracy that we use are provided in Appendix F.

For each method, we trained the network for a grid of initial step sizes with values

$$\left(10^{-4}, 5 \cdot 10^{-4}, 10^{-3}, 5 \cdot 10^{-3}, 10^{-2}, 5 \cdot 10^{-2}, 0.1, 0.5, 1\right),$$

upon which the step size yielding the best result on the test data was chosen.

Figure 4 displays the perplexity, averaged over the random seeds. Adam and SGD with momentum perform slightly better on the training data but also exhibit a higher tendency to overfit. The usage of the differentiable clipping functions appears to have a regularizing effect on this task. Besides these differences, the algorithms demonstrate comparable performances on the given problem.

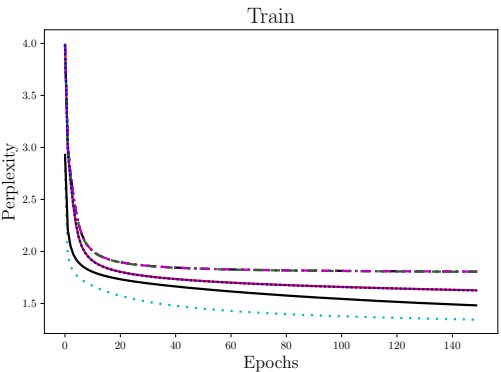
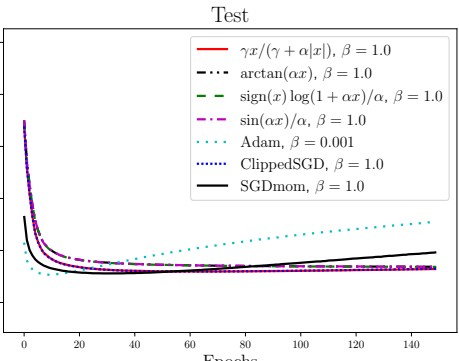

Figure 4: Perplexity of the different methods when used to train a one-layer recurrent neural network for next-character prediction on the Penn Treebank data set. The functions in the legend correspond to the component-wise clipping functions listed in Remark 3 and in Examples 1–4 in Appendix B. By 'Adam', 'SGDmom' and 'ClippedSGD' we denote the methods Adam, SGD with momentum and component-wise, clipped SGD respectively. For all the methods, $\beta > 0$ is the initial step size parameter.

## 7  CONCLUSIONS

We have analyzed and investigated the behavior of a large class of soft clipping algorithms. On common large scale machine learning tasks, we have seen that they exhibit similar performance to state of the art algorithms such as Adam and SGD with momentum.

In the strongly convex, case we proved $\mathcal{O}(\frac{1}{K})-$convergence of the iterates' function values to the value of the unique minimum. In the non-convex case, we demonstrated convergence to a stationary point at a rate of $\mathcal{O}(\frac{1}{\log(K)})$ in expectation, as well as almost sure convergence.

Overall, we see that the algorithms we have investigated exhibit a similar performance to state-of-the-art algorithms. In problems where other algorithms may display a tendency to overfit, the differentiability of the clipping functions used in the soft-clipping schemes may have a regularization effect. The analysis we have presented lays the theoretical foundations for the usage of a large class of stabilizing soft-clipping algorithms, as well as further research in the field.

## 8  REPRODUCIBILITY

All details necessary for reproducing the numerical experiments in this article are given in Section 6. The implementation of the schemes using the functions from Remark 3 or Appendix B is a straight-forward modification to the standard training loop in Tensorflow (Abadi et al., 2015).

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

# A    SETTING & ASSUMPTIONS

In the following, we consider the space $\mathbb{R}^d$, $d \in \mathbb{N}$. For $x = (x_1, \ldots, x_d)^T \in \mathbb{R}^d$, we denote the Euclidean norm by $\|x\| = (\sum_{i=1}^d x_i^2)^{\frac{1}{2}}$. Further, we denote the positive real numbers by $\mathbb{R}^+$ and the non-negative real numbers by $\mathbb{R}_0^+$. Let $(\Omega, \mathcal{F}, \mathbb{P})$ be a complete probability space and let $\{\xi_k\}_{k \in \mathbb{N}}$ be a family of jointly independent random variables on $\Omega$. We use the notation $\mathbb{E}_{\xi_k}[X]$ for the conditional expectation $\mathbb{E}[X | \sigma(\xi_1, \ldots, \xi_{k-1})]$ where $\sigma(\xi_1, \ldots, \xi_{k-1}) \subseteq \mathcal{F}$ is the $\sigma-$algebra generated by $\xi_1, \ldots, \xi_{k-1}$. By the tower property of the conditional expectation it holds that

$$\mathbb{E}\left[\mathbb{E}_{\xi_k}[X]\right] = \mathbb{E}[X].$$

From the mutual independence of the $\{\xi_k\}_{k \in \mathbb{N}}$ it also follows that the joint distribution factorizes to the product of the individual distributions.

We make the following assumptions on the methods:

**Assumption 1.** *Let the functions $G \colon \mathbb{R}^d \times \mathbb{R}^+ \to \mathbb{R}^d$ and $H \colon \mathbb{R}^d \times \mathbb{R}^+ \to \mathbb{R}^d$ fulfill the following properties:*

  *1. There exists $c_g \in \mathbb{R}^+$ such that $\|G(x, \alpha)\| \leq c_g \|x\|$ for every $x \in \mathbb{R}^d$ and $\alpha \in \mathbb{R}^+$.*

  *2. There exists $c_h \in \mathbb{R}_0^+$ such that $\|H(x, \alpha)\| \leq c_h \|x\|^2$ for every $x \in \mathbb{R}^d$ and $\alpha \in \mathbb{R}^+$.*

**Remark 3.** *In the following, the example for $G$ and $H$ that we are most interested in is that the functions take on the form*

$$G((x_1, \ldots, x_d)^T, \alpha) = \left(g_1(x_1, \alpha), \ldots, g_d(x_d, \alpha)\right)^T,$$
$$H((x_1, \ldots, x_d)^T, \alpha) = \left(h_1(x_1, \alpha), \ldots, h_d(x_d, \alpha)\right)^T,$$

*such that the component functions $g_i$ and $h_i$ fulfill:*

  *1. There exists $c_g \in \mathbb{R}^+$ such that $|g_i(x_i, \alpha)| \leq c_g |x_i|$ for every $x_i \in \mathbb{R}$ and $\alpha \in \mathbb{R}^+$.*

  *2. There exists $c_h \in \mathbb{R}_0^+$ such that $|h_i(x_i, \alpha)| \leq c_h |x_i|^2$ for every $x_i \in \mathbb{R}$ and $\alpha \in \mathbb{R}^+$.*

*These conditions imply that Assumption 1 is fulfilled: First, we observe that for the function $G$, we see that*

$$\|G(x, \alpha_k)\| = \Big(\sum_{i=1}^d |g_i(x_i, \alpha_k)|^2\Big)^{\frac{1}{2}} \leq \Big(c_g^2 \sum_{i=1}^d |x_i|^2\Big)^{\frac{1}{2}} \leq c_g \|x\|.$$

*Moreover, applying the inequality $\left(\sum_{i=1}^d a_i\right)^{\frac{1}{2}} \leq \sum_{i=1}^d a_i^{\frac{1}{2}}$ for $a_i \geq 0$, $i \in \{1, \ldots, d\}$, for the function $H$, it follows that*

$$\|H(x, \xi_k), \alpha)\| = \Big(\sum_{i=1}^d |h_i(x_i, \alpha)|^2\Big)^{\frac{1}{2}} \leq c_h \sum_{i=1}^d |x_i|^2 = c_h \|x\|^2.$$

*This setting allows for many variants of methods. Under the assumption that $g_i(x_i, \alpha) = g(x_i, \alpha)$, a few examples include the specific functions $g(x_i, \alpha) = \frac{\gamma x_i}{\gamma + \alpha |x_i|}$, $g(x_i, \alpha) = \frac{1}{\alpha} \arctan(\alpha x_i)$, $g(x_i, \alpha) = \frac{1}{\alpha} \operatorname{sign}(x_i) \ln(1 + \alpha |x_i|)$ and $g(x_i, \alpha) = \frac{1}{\alpha} \sin(\alpha x_i)$. The corresponding functions $h$ and proofs of these assertions are given in Appendix B.*

We also require the following standard assumptions about the problem to be solved:

**Assumption 2** (Unbiased gradients). *There exists a function $\nabla f(\cdot, \xi) \colon \mathbb{R}^d \times \Omega \to \mathbb{R}^d$ which is an unbiased estimate of the gradient $\nabla F$ of the objective function, i.e. it holds that*

$$\mathbb{E}[\nabla f(v, \xi)] = \nabla F(v) \quad \text{for all } v \in \mathbb{R}^d.$$

**Assumption 3** (Lipschitz continuity of gradient). *The objective function $F \colon \mathbb{R}^d \to \mathbb{R}$ is continuously differentiable and its gradient $\nabla F \colon \mathbb{R}^d \to \mathbb{R}^d$, is Lipschitz continuous with Lipschitz constant $L \in \mathbb{R}^+$, i.e. it holds that*

$$\|\nabla F(v) - \nabla F(w)\| \leq L \|v - w\| \quad \text{for all } v, w \in \mathbb{R}^d.$$

*Moreover, the stochastic gradient $\nabla f(\cdot, \xi)$ from Assumption 2 is Lipschitz continuous with $\sigma(\xi)$-measurable Lipschitz constant $L_\xi \in L^2(\Omega; \mathbb{R}^+)$, i.e.,*

$$\|\nabla f(v, \xi) - \nabla f(w, \xi)\| \leq L_\xi \|v - w\| \quad \text{for all } v, w \in \mathbb{R}^d \text{ almost surely,}$$

*and $\mathbb{E}\left[L_\xi^2\right] \leq L^2$.*

**Remark 4.** *Assumption 3 implies the* L-smoothness *condition, this means that*

$$F(w) \leq F(v) + \langle \nabla F(v), w - v \rangle + \frac{L}{2}\|w - v\|^2, \quad \text{for all } v, w \in \mathbb{R}^d,$$

*is fulfilled. Compare Equation (4.3) and Appendix B in Bottou et al. (2018).*

In the convergence analysis, we need some knowledge about how the stochastic gradient behaves around a local minimum. A first possibility is to assume square integrability:

**Assumption 4** (Bounded variance). *There exists $w_* \in \arg\min_{w \in \mathbb{R}^d} F(w)$ such that*

$$\mathbb{E}\left[\|\nabla f(w_*, \xi)\|^2\right] \leq \sigma^2,$$

*i.e. the variance is bounded at a local minimum $w_*$.*

An alternative, stronger assumption is to ask that the stochastic gradient, just like the full gradient, is zero at a chosen local minimum:

**Assumption 5** ($\nabla f(w_*, \xi) = 0$). *For $w_* \in \arg\min_{w \in \mathbb{R}^d} F(w)$ it additionally holds that $w_* \in \arg\min_{w \in \mathbb{R}^d} f(w, \xi)$ almost surely. In particular, this implies that for this $w_*$ it follows that $\nabla f(w_*, \xi) = 0$ almost surely.*

Such an assumption also appears in e.g. Ma et al. (2018) and Gorbunov (2023). While Assumption 5 is certainly stronger than Assumption 4, it is still reasonable to assume this when considering applications in a machine learning setting. These models are frequently *over-parameterized*, i.e. the number of parameters of the model are much larger than the number of samples in the data set. It is not uncommon for models of the like to have the capability to *interpolate* the training data and achieve 0 loss, see e.g. Ma et al. (2018); Vaswani et al. (2019). If the model $m$ completely interpolates the data in a setting where $F$ is given by (3), there is a parameter configuration $w_*$ such that $m(x_i, w_*) = y_i$ for all $i$, and thus $\ell(m(x_i, w_*), y_i) = 0$ for all $i$. Specifically, this means that there is a point $w_*$ that is the minimum of all the functions $w \mapsto \ell(m(x_i, w), y_i)$ at the same time. Such models still generalize well to unseen data, see e.g. Ma et al. (2018); Neyshabur et al. (2019); Vaswani et al. (2019).

In the setting explained in the previous assumptions, we now consider the following method given by

$$\begin{aligned}
w_{k+1} &= w_k - \alpha_k G\big(\nabla f(w_k, \xi_k), \alpha_k\big) \\
&= w_k - \alpha_k \nabla f(w_k, \xi_k) + \alpha_k^2 H(\nabla f(w_k, \xi_k), \alpha_k).
\end{aligned} \tag{12}$$

For the sequence $(w_k)_{k \in \mathbb{N}}$ of increments given by the method (12), we make an additional stability assumption:

**Assumption 6** (Moment bound). *For the initial value $w_1 \in \mathbb{R}^d$ and the sequence $(w_k)_{k \in \mathbb{N}}$ defined through the method (12), there exists a $M \in \mathbb{R}^+$ such that $w_* \in \arg\min_{w \in \mathbb{R}^d} F(w)$ from Assumption 4 also satisfies*

$$\mathbb{E}\left[\|w_k - w_*\|^3\right] \leq M$$

*for every $k \in \mathbb{N}$.*

Such an a priori result was established for the tamed SGD in Eisenmann & Stillfjord (2022), and similar techniques could likely be used in this more general situation. An alternative approach would be to impose a step size restriction such as in, e.g., Bottou et al. (2018, Theorem 4.8). If it cannot be established a priori, it is easy to detect in practice if the assumption does not hold.

The following analysis will require us to handle also terms of the form $\mathbb{E}\left[\|w_k - w_*\|^q\right]$ with $q = 1$ and $q = 2$, but as the following lemma shows these are also bounded if Assumption 6 is fulfilled.

**Lemma 3.** *Let Assumption 6 be fulfilled. Then*

$$\mathbb{E}\big[\|w_k - w_*\|^q\big] \leq M^{q/3} < \infty$$

*for every $k \in \mathbb{N}$ and $q \in [1,3]$.*

The proof is by Hölder's inequality, see Appendix C.

## B   EXAMPLE METHODS

Here, we provide further details on a few methods which satisfy Assumption 1 and the setting explained in Remark 3.

**Example 1.** *The component-wise soft-clipping scheme* (8)*, given by*

$$g(x, \alpha) = \frac{\gamma x}{\gamma + \alpha |x|} \quad and \quad h(x, \alpha) = \frac{x|x|}{\gamma + \alpha |x|}$$

*satisfies Assumption 1, since*

$$|g(x, \alpha)| = \Big|\frac{\gamma x}{\gamma + \alpha |x|}\Big| \leq |x| \quad and \quad |h(x, \alpha)| = \frac{\gamma |x|^2}{\gamma + \alpha |x|} \leq |x|^2.$$

**Example 2.** *The component-wise* arctan *scheme, given by*

$$g(x, \alpha) = \frac{\arctan(\alpha x)}{\alpha} \quad and \quad h(x, \alpha) = \frac{x}{\alpha} - \frac{\arctan(\alpha x)}{\alpha^2}$$

*satisfies Assumption 1. This can be seen since $\arctan(x) \leq x$ immediately implies $|g(x, \alpha)| \leq |x|$. Moreover, by expanding* arctan *in a first-order Taylor series with remainder term,*

$$\arctan(x) = x - \frac{\varepsilon}{(1 + \varepsilon^2)^2} x^2,$$

*where $|\varepsilon| \leq x$. It is easily determined that $\big|\frac{\varepsilon}{(1+\varepsilon^2)^2}\big| \leq \frac{1}{3}$ independently of $\varepsilon$. This then shows that for $|\varepsilon| \leq x$,*

$$|h(x, \alpha)| = \Big|\frac{\varepsilon}{(1 + \varepsilon^2)^2}\Big| x^2 \leq \frac{1}{3} x^2.$$

**Example 3.** *The component-wise logarithmic scheme, given by*

$$g(x, \alpha) = \frac{\text{sign}(x) \ln(1 + \alpha |x|)}{\alpha} \quad and \quad h(x, \alpha) = \frac{x}{\alpha} - \frac{\text{sign}(x) \ln(1 + \alpha |x|)}{\alpha^2}$$

*satisfies Assumption 1. This again follows since $\ln(1 + |x|) \leq |x|$ directly shows that $|g(x, \alpha)| = \frac{\ln(1+\alpha|x|)}{\alpha} \leq |x|$. Moreover, by a second-order Taylor expansion with remainder term, we have*

$$\ln(1 + |x|) = |x| - \frac{1}{2}|x|^2 + \frac{1}{3(1 + \varepsilon)^3}|x|^3 > |x| - \frac{1}{2}|x|^2,$$

*for $\varepsilon \in (0, |x|)$. This means that*

$$x - \frac{1}{2}|x|^2 \leq \text{sign}(x) \ln(1 + |x|) \leq x + \frac{1}{2}|x|^2.$$

*Thus, we get*

$$|h(x, \alpha)| \leq \frac{1}{2} x^2.$$

Essentially, any function $g(x, \alpha)$ for which $\alpha g(x, \alpha)$ looks like $\alpha x$ for small $\alpha x$ and which is bounded for large $\alpha x$ satisfies the assumption. Thus we also have, e.g.,

**Example 4.** *The trigonometric component-wise scheme given by*

$$g(x, \alpha) = \frac{\sin(\alpha x)}{\alpha} \quad and \quad h(x, \alpha) = \frac{x}{\alpha} - \frac{\sin(\alpha x)}{\alpha^2}.$$

*satisfies Assumption 1. We immediately verify $|g(x, \alpha)| = |\frac{\sin(\alpha x)}{\alpha}| \leq |x|$. Moreover, expansion in Taylor series shows that $\sin(x) = x - \frac{\sin(\varepsilon)}{2} x^2$ with $\varepsilon \leq |x|$. Thus, it follows that*

$$|h(x, \alpha)| = \frac{1}{\alpha^2}\Big|\frac{\sin(\varepsilon)}{2}(\alpha x)^2\Big| \leq \frac{1}{2} x^2.$$

## C  PROOFS

**Lemma 3.** *Let Assumption 6 be fulfilled. Then*

$$\mathbb{E}\big[\|w_k - w_*\|^q\big] \leq M^{q/3} < \infty$$

*for every $k \in \mathbb{N}$ and $q \in [1, 3]$.*

*Proof.* The lemma follows by Hölder's inequality, since

$$\mathbb{E}\big[\|w_k - w_*\|^q\big] = \int_\Omega \|w_k - w_*\|^q \mathrm{d}\mathbb{P}$$

$$= \Big( \int_\Omega \|w_k - w_*\|^3 \mathrm{d}\mathbb{P}\Big)^{q/3} \Big( \int_\Omega 1^{\frac{1}{1-q/3}} \mathrm{d}\mathbb{P}\Big)^{1-q/3}$$

$$= \mathbb{E}\big[\|w_k - w_*\|^3\big]^{q/3} \cdot 1.$$

$\square$

**Lemma 1.** *Let Assumptions 1, 2, 3, 4 and 6 be fulfilled. Further, let $\{w_k\}_{k \in \mathbb{N}}$ be the sequence generated by the method (12). Then it holds that*

$$\mathbb{E}\big[F(w_{k+1})\big] - \mathbb{E}\big[F(w_k)\big] \leq -\alpha_k \mathbb{E}\big[\|\nabla F(w_k)\|^2\big] + \alpha_k^2 B_1,$$

*where*

$$B_1 = 2c_h L^3 M + 2c_h L M^{1/3}\sigma^2 + c_g^2 L^3 M^{2/3} + c_g^2 L\sigma^2.$$

*Proof.* First, we apply Assumptions 1 and 3 as well as Remark 4, to obtain

$$F(w_{k+1}) - F(w_k) \leq \langle \nabla F(w_k), w_{k+1} - w_k \rangle + \frac{L}{2}\|w_{k+1} - w_k\|^2$$

$$= -\alpha_k \langle \nabla F(w_k), \nabla f(w_k, \xi_k)\rangle$$

$$\quad + \alpha_k^2 \langle \nabla F(w_k), H\left(\nabla f(w_k, \xi_k), \alpha_k\right)\rangle \tag{13}$$

$$\quad + \alpha_k^2 \frac{L}{2}\|G\left(\nabla f(w_k, \xi_k), \alpha_k\right)\|^2. \tag{14}$$

Now we take $w_*$ such that $\nabla F(w_*) = 0$. From Assumption 1, it follows that

$$\|H\left(\nabla f(w_k, \xi_k), \alpha\right)\| \leq c_h \|\nabla f(w_k, \xi_k)\|^2.$$

By the Cauchy-Schwarz inequality and Assumption 3 we can therefore bound (13) as

$$\alpha_k^2 \langle \nabla F(w_k), H\left(\nabla f(w_k, \xi_k), \alpha_k\right)\rangle \leq \alpha_k^2 \|\nabla F(w_k) - \nabla F(w_*)\| \|H\left(\nabla f(w_k, \xi_k), \alpha_k\right)\|$$

$$\leq \alpha_k^2 c_h L \|w_k - w_*\| \|\nabla f(w_k, \xi_k)\|^2.$$

Applying Assumption 3 and 4, we find

$$\mathbb{E}_{\xi_k}\big[\|\nabla f(w_k, \xi_k)\|^2\big] \leq 2\mathbb{E}_{\xi_k}\big[\|\nabla f(w_k, \xi_k) - \nabla f(w_*, \xi_k)\|^2\big] + 2\mathbb{E}_{\xi_k}\big[\|\nabla f(w_*, \xi_k)\|^2\big]$$

$$\leq 2\mathbb{E}_{\xi_k}\big[L_{\xi_k}^2\big]\|w_k - w_*\|^2 + 2\sigma^2,$$

since $\|w_k - w_*\|^2$ is stochastically independent of $\xi_k$. Since $\mathbb{E}_{\xi_k}\big[L_{\xi_k}^2\big] \leq L^2$, we find that

$$\mathbb{E}_{\xi_k}\big[\alpha_k^2 \langle \nabla F(w_k), H\left(\nabla f(w_k, \xi_k), \alpha_k\right)\rangle\big]$$

$$\leq 2\alpha_k^2 c_h L^3 \|w_k - w_*\|^3 + 2\alpha_k^2 c_h L\|w_k - w_*\|\sigma^2.$$

Moreover, due to Assumption 4, we can bound (14) as

$$\frac{L}{2}\mathbb{E}_{\xi_k}\big[\|G\left(\nabla f(w_k, \xi_k), \alpha_k\right)\|^2\big]$$

$$\leq \frac{c_g^2 L}{2}\mathbb{E}_{\xi_k}\big[\|\nabla f(w_k, \xi_k)\|^2\big]$$

$$\leq c_g^2 L \mathbb{E}_{\xi_k}\big[\|\nabla f(w_k, \xi_k) - \nabla f(w_*, \xi_k)\|^2\big] + c_g^2 L \mathbb{E}_{\xi_k}\big[\|\nabla f(w_*, \xi_k)\|^2\big]$$

$$\leq c_g^2 L \mathbb{E}_{\xi_k}\big[L_{\xi_k}^2\big]\|w_k - w_*\|^2 + c_g^2 L\sigma^2$$

$$\leq c_g^2 L^3 \|w_k - w_*\|^2 + c_g^2 L\sigma^2.$$

By Assumption 2 and Equation (10.17) in Resnick (2014) it holds that

$$\mathbb{E}_{\xi_k}\left[\nabla f(w_k, \xi_k)\right] = \nabla F(w_k),$$

where we have used the fact that the variables $\{\xi_k\}_{k\in\mathbb{N}}$ are independent and that $w_k$ only depends on $\xi_j$ for $j \leq k-1$. Combining the bounds for (13) and (14) and taking the conditional expectation then leads to

$$\begin{aligned}
\mathbb{E}_{\xi_k}\left[F(w_{k+1})\right] - F(w_k) \leq\ & -\alpha_k\|\nabla F(w_k)\|^2 \\
& + 2\alpha_k^2 c_h\big(L^3\|w_k - w_*\|^3 + L\|w_k - w_*\|\sigma^2\big) \\
& + \alpha_k^2 c_g^2\big(L^3\|w_k - w_*\|^2 + L\sigma^2\big).
\end{aligned}$$

Taking the expectation and making use of Assumption 6 and Lemma 3, we then obtain the claimed bound. $\qquad\square$

**Lemma 2.** *Let Assumptions 1, 2, 3, 5 and 6 be fulfilled. Further, let $\{w_k\}_{k\in\mathbb{N}}$ be the sequence generated by the method (12). Then it holds that*

$$\mathbb{E}\left[F(w_{k+1})\right] - \mathbb{E}\left[F(w_k)\right] \leq -\alpha_k\mathbb{E}\left[\|\nabla F(w_k)\|^2\right] + \alpha_k^2 B_2,$$

*where*

$$B_2 = c_h L^3 M + \frac{c_g^2 L^3}{2}M^{2/3}.$$

*Proof.* Analogously to the proof of Lemma 1, we find that

$$\begin{aligned}
F(w_{k+1}) - F(w_k) \leq\ & -\alpha_k\langle\nabla F(w_k), \nabla f(w_k, \xi_k)\rangle \\
& + \alpha_k^2\Big(c_h L\|w_k - w_*\| + \frac{c_g^2 L}{2}\Big)\|\nabla f(w_k, \xi_k)\|^2.
\end{aligned}$$

But by Assumption 3 and 5, it follows that

$$\mathbb{E}_{\xi_k}\left[\|\nabla f(w_k, \xi_k)\|^2\right] \leq \mathbb{E}_{\xi_k}\left[L_{\xi_k}^2\right]\|w_k - w_*\|^2 \leq L^2\|w_k - w_*\|^2.$$

After taking the conditional expectation and applying the fact that $\mathbb{E}_{\xi_k}\left[\nabla f(w_k, \xi_k)\right] = \nabla F(w_k)$, we obtain

$$\begin{aligned}
\mathbb{E}_{\xi_k}\left[F(w_{k+1})\right] - F(w_k) \leq\ & -\alpha_k\|\nabla F(w_k)\|^2 \\
& + \alpha_k^2 c_h L^3\|w_k - w_*\|^3 + \frac{c_g^2 L^3}{2}\|w_k - w_*\|^2.
\end{aligned}$$

Taking the expectation and making use of Assumption 6 and Lemma 3, we obtain the claimed bound. $\qquad\square$

**Theorem 1.** *Let Assumptions 1, 2, 3 and 6 as well as Assumption 4 or Assumption 5 be fulfilled. Further, let $\{w_k\}_{k\in\mathbb{N}}$ be the sequence generated by the method (12). Then it follows that*

$$\min_{1\leq k\leq K}\mathbb{E}\left[\|\nabla F(w_k)\|^2\right] \leq \frac{F(w_1) - F(w_*)}{\sum_{k=1}^K \alpha_k} + B_i\frac{\sum_{k=1}^K \alpha_k^2}{\sum_{k=1}^K \alpha_k}, \qquad (10)$$

*where the constant $B_i$, $i \in \{1, 2\}$, is stated in Lemma 1 (with Assumption 4) and Lemma 2 (with Assumption 5), respectively.*

*Under the standard assumption that $\sum_{k=1}^\infty \alpha_k^2 < \infty$ and $\sum_{k=1}^\infty \alpha_k = \infty$, in particular, it holds that*

$$\lim_{K\to\infty}\min_{1\leq k\leq K}\mathbb{E}\left[\|\nabla F(w_k)\|^2\right] = 0.$$

*Proof.* By assumption, we have that

$$\mathbb{E}\left[F(w_{k+1})\right] - \mathbb{E}\left[F(w_k)\right] \leq -\alpha_k\mathbb{E}\left[\|\nabla F(w_k)\|^2\right] + \alpha_k^2 B_i,$$

which we rearrange to

$$\alpha_k\mathbb{E}\left[\|\nabla F(w_k)\|^2\right] \leq \mathbb{E}[F(w_k)] - \mathbb{E}\left[F(w_{k+1})\right] + \alpha_k^2 B_i.$$

Summing from $k = 1$ to $K$ now gives

$$\sum_{k=1}^{K} \alpha_k \mathbb{E}\left[\|\nabla F(w_k)\|^2\right] \leq F(w_1) - F(w_*) + B_i \sum_{k=1}^{K} \alpha_k^2.$$

where we have used the fact that $\mathbb{E}\left[F(w_{K+1})\right] \geq F(w_*)$ and that $\mathbb{E}\left[F(w_1)\right] = F(w_1)$. It then follows that

$$\min_{1 \leq k \leq K} \mathbb{E}\left[\|\nabla F(w_k)\|^2\right] \leq \frac{F(w_1) - F(w_*)}{\sum_{k=1}^{K} \alpha_k} + B_i \frac{\sum_{k=1}^{K} \alpha_k^2}{\sum_{k=1}^{K} \alpha_k},$$

which tends to 0 as $K \to \infty$. $\qquad\square$

**Corollary 1.** *Let the conditions of Theorem 1 be fulfilled and suppose that the step size sequence in (12) is defined by $\alpha_k = \frac{\beta}{k+\gamma}$. Then it follows that*

$$\min_{1 \leq k \leq K} \mathbb{E}\left[\|\nabla F(w_k)\|^2\right] \leq \frac{F(w_1) - F(w_*)}{\beta \ln(K + \gamma + 1)} + B_i \frac{\beta(2 + \gamma)}{\ln(K + \gamma + 1)},$$

*where the constant $B_i$, $i \in \{1, 2\}$, is stated in Lemma 1 (with Assumption 4) and Lemma 2 (with Assumption 5), respectively.*

*Proof.* This statement follows from Theorem 1 and the following integral estimates of the appearing sums:

$$\sum_{k=1}^{K} \alpha_k^2 \leq \int_1^K \frac{\beta^2}{(x + \gamma)^2} \mathrm{d}x + \frac{\beta^2}{(1 + \gamma)^2} \leq \beta^2 \frac{2 + \gamma}{(1 + \gamma)^2}$$

and

$$\sum_{k=1}^{K} \alpha_k \geq \int_1^{K+1} \frac{\beta}{x + \gamma} \mathrm{d}x = \beta \ln(K + \gamma + 1).$$

$\qquad\square$

**Corollary 3.** *Let the conditions of Theorem 1 be fulfilled. It follows that the sequence*

$$\{\zeta_K(\omega)\}_{K \in \mathbb{N}}, \text{ where } \zeta_K(\omega) = \min_{1 \leq k \leq K} \|\nabla F(w_k(\omega))\|_2^2, \tag{11}$$

*converges to 0 for almost all $\omega \in \Omega$, i.e.*

$$\mathbb{P}\left(\left\{\omega \in \Omega : \lim_{K \to \infty} \zeta_K(\omega) = 0\right\}\right) = 1.$$

*Proof.* By (10), the sequence

$$\left\{\zeta_K(\omega)\right\}_{K \in \mathbb{N}} \quad \text{with} \quad \zeta_K(\omega) = \min_{1 \leq k \leq K} \|\nabla F(w_k(\omega))\|_2^2$$

converges in expectation to 0 as $K \to \infty$. Since convergence in expectation implies convergence in probability (see Cohn (2013, Prop. 3.1.5)), for every $\varepsilon > 0$ it holds that

$$\lim_{K \to \infty} \mathbb{P}\left(\{\omega \in \Omega : \zeta_K(\omega) > \varepsilon\}\right) = 0. \tag{15}$$

Furthermore, the sequence is decreasing; i.e. for every $K \in \mathbb{N}$ we have that $\zeta_{K+1}(\omega) \leq \zeta_K(\omega)$ almost surely. Hence

$$\sup_{k \geq K} \zeta_k = \zeta_K, \quad \text{for all } K \in \mathbb{N}. \tag{16}$$

A standard result in probability theory (compare Shiryaev (2016, Thm. 1, Sec. 2.10.2)) states that a sequence $\{\zeta_K\}_{K \in \mathbb{N}}$ converges a.s. to a random variable $\zeta$ if and only if

$$\mathbb{P}\left(\left\{\omega \in \Omega : \sup_{k \geq K} |\zeta_k(\omega) - \zeta(\omega)| \geq \varepsilon\right\}\right) = 0, \tag{17}$$

for every $\varepsilon > 0$. Combining (15) and (16) we see that (17) holds for $\{\zeta_K(\omega)\}_{K \in \mathbb{N}}$ with $\zeta = 0$. $\quad\square$

**Theorem 2.** *Let Assumptions 1, 2, 3 and 6 as well as Assumption 4 or Assumption 5 be fulfilled. Additionally, let $F$ be strongly convex with convexity constant $c \in \mathbb{R}^+$, i.e.*

$$\langle \nabla F(v) - \nabla F(w), v - w \rangle \geq c \|v - w\|^2$$

*is fulfilled for all $v, w \in \mathbb{R}^d$. Further, let $\alpha_k = \frac{\beta}{k+\gamma}$ with $\beta, \gamma \in \mathbb{R}^+$ such that $\beta \in (\frac{1}{2c}, \frac{1+\gamma}{2c})$. Finally, let $\{w_k\}_{k \in \mathbb{N}}$ be the sequence generated by the method (12). Then it follows that*

$$F(w_k) - F(w_*) \leq \frac{(1+\gamma)^{2\beta c}}{(k+1+\gamma)^{2\beta c}} \big( F(w_1) - F(w_*) \big) + \frac{B_i \mathrm{e}^{\frac{2\beta c}{1+\gamma}}}{2\beta c - 1} \cdot \frac{1}{k+1+\gamma},$$

*where the constant $B_i$, $i \in \{1, 2\}$, is stated in Lemma 1 (with Assumption 4) and Lemma 2 (with Assumption 5), respectively.*

*Proof.* From Lemma 1 and Lemma 2 we get

$$\mathbb{E}\big[F(w_{k+1})\big] - \mathbb{E}\big[F(w_k)\big] \leq -\alpha_k \mathbb{E}\big[\|\nabla F(w_k)\|^2\big] + \alpha_k^2 B_i.$$

Since strong convexity implies that $2c(F(w_k) - F(w_*)) \leq \|\nabla F(w_k)\|^2$, see e.g. Inequality (4.12) in Bottou et al. (2018), this is equivalent to

$$\mathbb{E}\big[F(w_{k+1})\big] - F(w_*) \leq (1 - 2c\alpha_k)\big(\mathbb{E}\big[F(w_k)\big] - F(w_*)\big) + \alpha_k^2 B_i.$$

Iterating this inequality leads to

$$\mathbb{E}\big[F(w_{k+1})\big] - F(w_*) \leq \prod_{j=1}^{k} (1 - 2c\alpha_j)\big(\mathbb{E}\big[F(w_1)\big] - F(w_*)\big)$$

$$+ B_i \sum_{j=1}^{k} \alpha_j^2 \prod_{i=j+1}^{k} (1 - 2c\alpha_i).$$

With $\alpha_k = \frac{\beta}{k+\gamma}$ and the given bounds on $\beta$ and $\gamma$, we can now apply Lemma A.1 from Eisenmann & Stillfjord (2022) with $x = 2\beta c$ and $y = \gamma$ to bound the product and sum-product terms. This results in the claimed bound. $\square$

## D    EXPERIMENT 1 DETAILS

The cost functional in the first numerical experiment in Section 6.1 is

$$F(w) = \frac{1}{N} \sum_{i=1}^{N} f(x^i, w) = \frac{1}{N} \sum_{i=1}^{N} \sum_{j=1}^{d} \frac{(x_j^i)^2 w_j^2 + 13}{d},$$

where $N = 1000$, $d = 50$, and the vector $w \in \mathbb{R}^d$ contains the optimization parameters. Each $x^i \in \mathbb{R}^d$ is a known data vector which was sampled randomly from normal distributions with standard deviation 1 and mean $1 + \frac{10i}{d}$. This means that

$$\nabla F(w) = Aw + b,$$

where $A$ is a diagonal matrix with the diagonal entries

$$\lambda_j = A_{j,j} = \frac{2}{Nd} \sum_{i=1}^{N} (x_j^i)^2$$

and $b$ is a vector with the components

$$b_j = \frac{2 \cdot 13}{Nd} \sum_{i=1}^{N} x_j^i.$$

Further, we approximate $\nabla F$ using a batch size of 32, i.e.

$$\nabla f(w_k, \xi_k) = \frac{1}{|B_{\xi_k}|} \sum_{i \in B_{\xi_k}} \nabla f(x^i, w_k)$$

where $B_{\xi_k} \subset \{1, \ldots, N\}$ with $|B_{\xi_k}| = 32$. Similarly to $\nabla F$, this means that we can write the approximation as

$$\nabla f(w_k, \xi_k) = \tilde{A}(\xi_k)w_k + \tilde{b}(\xi_k),$$

where $\tilde{A}(\xi_k)$ is a diagonal matrix with the diagonal entries

$$\tilde{\lambda}_j(\xi_k) = \tilde{A}(\xi_k)_{j,j} = \frac{2}{|B_{\xi_k}|d} \sum_{i \in B_\xi} (x_j^i)^2,$$

and where $\tilde{b}(\xi_k)$ is a vector with components

$$\tilde{b}_j = \frac{2 \cdot 13}{|B_{\xi_k}|d} \sum_{i \in B_\xi}^N x_j^i.$$

## E    EXPERIMENT 2 DETAILS

The network used in the second experiment in Section 6.2 is a VGG network. This is a more complex type of convolutional neural network, first proposed in (Simonyan & Zisserman, 2015). Our particular network consists of three blocks, where each block consists of two convolutional layers followed by a $2 \times 2$ max-pooling layer and a dropout layer. The first block has a kernel size of $32 \times 32$, the second $64 \times 64$ and the last $128 \times 128$. The dropout percentages are 20, 30 and 40%, respectively. The final part of the network is a fully connected dense layer with 128 neurons, followed by another 20% dropout layer and an output layer with 10 neurons. The activation function is ReLu for the first dense layer and softmax for the output layer. We use a crossentropy loss function. The total network has roughly 550 000 trainable weights.

The dataset CIFAR-10 is a standard dataset from the Canadian institute for advanced research, consisting of 60000 32x32 colour images in 10 classes (Krizhevsky, 2009). We preprocess it by rescaling the data such that each feature has mean 0 and variance 1. During training, we also randomly flip each image horizontally with probability 0.5.

## F    EXPERIMENT 3 DETAILS

In the third experiment in Section 6.3 we consider the Pennsylvania Treebank portion of the Wall Street Journal corpus (Marcus et al., 1993). Sections 0-20 of the corpus are used in the training set (around 5M characters) and sections 21-22 is used in the test set (around 400K characters). Since the vocabulary consists of 52 characters, this is essentially a classification problem with 52 classes. We use a simple recurrent neural network consisting of one embedding layer with 256 units, followed by an LSTM-layer of 1024 hidden units and a dense layer with 52 units (the vocabulary size). We use a 20% drop out on the input weight matrices. A categorical crossentropy loss function is used after having passed the output through a softmax layer.

It is common to measure the performance of language models by monitoring the *perplexity*. This is the exponentiated *averaged regret*, i.e.

$$\exp\left(\frac{1}{T}\sum_{k=1}^T f(w_k, \xi_k)\right),$$

where $T$ is the number of batches in an epoch. For a model that has not learned anything and at each step assigns a uniform probability to all the characters of the vocabulary, we expect the perplexity to be equal to the size of the vocabulary. In this case, 52. For a model that always assigns the probability 1 to the right character it should be equal to 1. See e.g. Graves (2013); Mikolov et al. (2011). In the experiments, we use a sequence length of 70 characters, similar to Merity et al. (2018); Zhang et al. (2020a). As in the first experiment, we train the network for 150 epochs with each method for 5 different seeds ranging from 0 to 4 and compute the mean perplexity for the training- and test sets.

