# OpenReview forum: "Analysis of a class of stochastic component-wise soft-clipping schemes"
_ICLR.cc/2024/Conference — Submitted to ICLR 2024_

### Official Review · Reviewer_GBzB · 2023-10-28

**Soundness:** 2 fair
**Presentation:** 2 fair
**Contribution:** 1 poor
**Rating:** 3
**Confidence:** 3

**Summary:**

This paper proposes and studies a class of component-wise soft clipping algorithms. For strongly convex and non-convex smooth objective functions, the authors establish convergence guarantees of the proposed methods to minimizer/ stationary points. To verify the theoretical results, the authors conduct experiments demonstrating the effectiveness of the soft clipping methods.

**Strengths:**

The paper is well written and is easy to read.

The theoretical analysis looks correct, though I do not check all the technical details. The soft clipping framework seems novel.

**Weaknesses:**

1) Although the authors mention in the abstract that theoretical guarantees can provide guidelines for us to choose what algorithms to use, there is little discussion about this aspect in the paper. More specifically, it is unclear through the theoretical results why or when soft clipping is better than hard clipping or vanilla SGD. For the non-convex case, the convergence rate is only $O(1/\log K)$ which is much slower that standard results for GD/SGD or (hard) gradient clipping, though the assumptions seem to be a little different.

2) It seems to me that although soft and hard clipping are not exactly equivalent, they are basically the same since the induced step size only differs by a constant scale. In the introduction, the authors motivate the study of soft clipping by saying that "A drawback of the rescaling in (4) is that it is not a differentiable function of the gradient", which looks confusing, since it is unclear why being differentiable here is favorable.

3) I think there are some existing works on either component-wise or non component-wise clipping, and some of them are cited in this paper, but there is little comparison between these papers and the current paper.

**Questions:**

1) In the experiments, do you compare component-wise clipped SGD with the original clipping method (non component wise)?

2) Can you briefly discuss why the theoretical results in this paper imply that component-wise clipped SGD is superior to other popular methods?

---

> ### Author Response · Authors · 2023-11-17
> **Response to GBzB**
>
> We thank the referee for the feedback.
>
> On the weaknesses comments:
>
> 1. With "a theoretical foundation" and "motivates their usage", we only intended to convey that we have proved that they converge under reasonable conditions, not that theory necessarily recommends a specific method. In fact, like we mentioned in the responses to the other referees, we do not yet have a full understanding of which method would be the optimal choice in a given situation. The benefit of soft clipping over hard clipping is mostly empirical, see e.g. the comments in [Zhang et al., 2020a]. As to why clipping should be preferred over vanilla SGD, see e.g.\ the references [3] and [4] provided by the referee xFTp. These should obviously have been referenced in our paper already and have been added to the revised version.
>
> 2. That hard and soft clipping are "basically the same" is essentially the argument in the paper by Zhang et al. (2020a), which has been propagated from a similar argument in previous papers with some of the same authors. This seems plausible when stated in this vague form, but when doing the details it becomes apparent that it is in fact not true. That is, the convergence proof for hard clipping does not translate to a proof of convergence for soft clipping. We have now added a few sentences commenting on this.
>
> As stated above, the fact that soft clipping is preferable to hard clipping is mostly empirical and we do not have a rigorous proof of why differentiability is desirable. However, experience from control theory and ODE time-stepping strongly suggests that this is the case. Essentially, it makes the errors more predictable, and the method more robust.
>
> 3. In the revised version, we have improved the sections on contributions and literature overview. This includes more comparisons between our work and that in related papers.
>
>
> On the Questions:
>
> 1. As stated in the text, we are comparing to the component-wise clipped SGD. We thought this was most relevant.
>
> 2. We do not claim that these methods are superior, and the analysis does not suggest this. It does, however, prove convergence for a general class of methods in a unified framework. In particular, we observe the somewhat unexpected result that the trigonometric rescaling converges. This doesn't mean that we expect it to be a great method. In fact, we included it mostly to show that the analysis is applicable to a large class of functions.

---

### Official Review · Reviewer_xFTp · 2023-11-01

**Soundness:** 3 good
**Presentation:** 3 good
**Contribution:** 1 poor
**Rating:** 3
**Confidence:** 4

**Summary:**

The paper considers the family of SGD-type methods with non-linear transformation of the stochastic gradient such that this transformation can be seen as a linear approximation of the stochastic gradient w.r.t. the stepsize. Under standard assumptions (Lipshitz gradients, bounded variance) and some non-standard assumptions (boundedness of the third moments of the iterates) the authors derive $\mathcal{O}(\frac{1}{\ln(K)})$ and $\mathcal{O}(\frac{1}{K})$ convergence rates for non-convex and strongly convex cases respectively. Some special cases of the methods fitting the considered framework are tested in the numerical experiments with the training of neural networks. They show similar (slightly worse sometimes) performances to the standard methods like Clipped-SGD or Adam.

**Strengths:**

S1. The paper is clearly written and the overall idea of considering different non-linearities/clipping operators is promising.

S2. The proofs are correct.

**Weaknesses:**

W1. The paper does not provide a justification of why and when it is beneficial to use the considered methods.

- W1.1. From the theoretical perspective, the existing results under considered assumptions are either stronger or match the derived ones. Indeed, under bounded variance and smoothness assumptions, standard SGD also converges (and the known rates are even better), e.g., see [1, 2].

- W1.2. From a practical perspective, the considered methods do not show better performance than the standard methods in the experiments considered in the paper. Moreover, since the stepsize decreases over time, all the considered methods will be very close to standard SGD (in view of formula (9)), which is known to have bad convergence on some problems with a lack of smoothness [3] or with a presence of the heavy-tailed noise [4, 5].

W2. Although the paper focuses on the theoretical analysis of the considered methods, the derived results require improvements.

- W2.1. Assumption 6 is quite restrictive: it is unclear apriori whether it holds in the considered setup. Indeed, it can be the case that $\mathbb{E}[\|\|w^k - w^\ast \|\|^3] = \infty$ while Assumptions 3 and 4 are satisfied, e.g., for simple quadratic function with linear noise $f(v,\xi) = \|\|v\|\|^2 + \langle \xi, v \rangle$, where $\xi$ is zero mean random vector with a bounded variance but an unbounded third moment (e.g., one can take shifted Pareto distribution with tails decaying as $x^{-4}$) we have Assumption 3 with $L_{\xi} \equiv 2$, Assumption 4 is also satisfied with some $\sigma$, but Assumption 6 is not satisfied. The authors refer to the work of Eisenmann & Stillfjord (2022), but in that work, it is assumed that the fourth moment of the gradient is bounded (and the fourth moment of $L_\xi$) to derive Assumption 6. Such assumptions are even stronger than for the standard analysis of SGD.

- W2.2. The result of Theorem 1 seems to be not tight: when full gradients are used the theorem does not recover the rate of a deterministic counterpart -- Gradient Descent. This happens because $B_i \neq 0$ even when full gradients are used in the method. A similar issue is present in Theorem 2. Moreover, one can improve the rate from Corollary 1 to $\mathcal{O}(1/\sqrt{K})$ (instead of $\mathcal{O}(\frac{1}{\ln(K)})$) if one takes $\alpha_k \equiv 1/\sqrt{K}$.

- W2.3. Assumption 1 does not include some interesting special cases as standard gradient clipping (the second part of the assumption does not hold).


[1] Ghadimi, S., & Lan, G. (2013). Stochastic first-and zeroth-order methods for nonconvex stochastic programming. SIAM Journal on Optimization, 23(4), 2341-2368.

[2] Gower, R. M., Loizou, N., Qian, X., Sailanbayev, A., Shulgin, E., & Richtárik, P. (2019, May). SGD: General analysis and improved rates. In International conference on machine learning (pp. 5200-5209). PMLR.

[3] Zhang, J., He, T., Sra, S., & Jadbabaie, A. (2019). Why gradient clipping accelerates training: A theoretical justification for adaptivity. arXiv preprint arXiv:1905.11881.

[4] Zhang, J., Karimireddy, S. P., Veit, A., Kim, S., Reddi, S., Kumar, S., & Sra, S. (2020). Why are adaptive methods good for attention models?. Advances in Neural Information Processing Systems, 33, 15383-15393.

[5] Sadiev, A., Danilova, M., Gorbunov, E., Horváth, S., Gidel, G., Dvurechensky, P., Gasnikov, A., & Richtárik, P. (2023). High-probability bounds for stochastic optimization and variational inequalities: the case of unbounded variance. arXiv preprint arXiv:2302.00999.

**Questions:**

My main question is about the motivation of this work: why and when SGD-type methods with soft clipping should be used? In the current shape, the theory shows no benefit over the existing results. Moreover, the derived results are even weaker since they rely on an additional restrictive assumption. The numerical experiments also do not provide enough evidence that the methods with soft-clipping can outperform the existing algorithms (in both experiments standard Clipped-SGD performs better than other methods with soft clipping).

---

> ### Author Response · Authors · 2023-11-17
> **Response to xFTp**
>
> We thank the referee for the feedback.
>
> On the Weaknesses comments:
>
> W.1.1: We might be missing something here, but as far as we can see, our rates are the same as for SGD. The result in [2] for strongly convex problems is $\mathcal{O}(1/k)$ which is the same as in our Corollary 2 (3 in the revised version). The main result Theorem 2.1 in [1] says that $\|\nabla f(w)\| = \mathcal{O}(1 / \sum \alpha_k)$, where $\alpha_k = 1/k$ would give a $\log k$ term in the denominator. One main difference is that the results for SGD require a step size restriction, which can be hard to evaluate a priori in practice. The clipped schemes do not have such a restriction.
>
> W1.2: Similar to our responses to the other referees, we agree that the methods do not outperform the methods we compare them to. But viewed from the other side, those methods also do not outperform the generalized clipped methods, which we think is quite significant. The main point of the paper is not to suggest a new "optimal" method, but to analyze this large general class of methods in a unified way. One of the main motivations for clipped schemes is precisely that they perform well in the case of heavy-tailed noise, as noted in [4] (the referee's reference, not our paper's [4]).
>
> W2.1: We agree that the assumption is somewhat non-standard and may not be fulfilled in every problem. But in practice, when there is only a finite amount of data and thus also finite possibilities for choosing the stochastic approximation, the third moment will be bounded if the second moment is. The setting in Eisenmann & Stillfjord (2022) indeed does not perfectly match our setting, in that it requires strong convexity. We mainly wanted to state an example included in our setting where such a moment bound on the increments can be observed. Note that the fourth moment of the approximate gradient is not assumed to be bounded for the basic convergence result, it is only used to improve the rate
>
>
> W2.2 (first comment): The price one pays for the generality is the larger error constant in the bound. In our case, the non-linearity of the clipping functions leads to an increased error constant. In practice, however, as the experiments show, it is not an issue.
>
> W2.2 (second comment): Yes, this is true. Our emphasis in the article is on the decreasing step size regime, but a slight extension yields $\mathcal{O}(1/\sqrt{K})$-convergence in the non-convex case with the mentioned step size. We have added a second corollary about this in the non-convex part, and a similar remark in the convex case.
>
>
> W2.3 We have updated our main assumption on the methods in line with referee pggZ's comments, and in the new version this method is included.
>
>
> On the comments in Questions:
>
> As stated in the responses to the other referees, we do not agree that the Clipped-SGD outperforms the other methods. They essentially perform equally well. We have tried to state more clearly now that the point of the paper is not to suggest new methods that always outperform the state of the art.

---

> > ### Comment · Reviewer_xFTp · 2023-11-23
> > **Response to the rebuttal**
> >
> > I thank the authors for their responses. However, my main concerns remain.
> >
> > Re W.1.1. Yes, my point was about this: standard results without clipping also provide similar guarantees. Moreover, some assumptions (like Assumption 6) are not needed in the analysis of SGD. The authors point out that the results for SGD require steps size restriction, while the clipped schemes that they consider do not require such restrictions. This happens due to the change of the assumptions that authors use: in the descent, Lemma 1, one can get rid of the extra squared gradient norm (appearing in the standard analysis of SGD) using Assumption 6. In fact, if we take $H \equiv 0$ the method becomes standard SGD, and the analysis still holds, i.e., for SGD one can also get no restrictions on the step size in the considered settings.
> >
> > Re W1.2. Since the proposed method can be seen as SGD with quadratic w.r.t. the stepsize perturbation, it is not obvious beforehand that these methods will behave similarly to the method with standard clipping on the heavy-tailed task. Moreover, due to the same reason, it is expected that the methods considered in the submission will behave similarly to SGD that is significantly outperformed by ClippedSGD and Adam in such tasks as considered in [4]. It is also worth mentioning that the settings considered in the submission do not allow heavy-tailed noise since they are stronger than the bounded variance assumption.
> >
> > Re W2.1. Even if the third moment exists (like in the finite-sum case), it is quite restrictive to assume beforehand that there exists some $M$ such that Assumption 6 holds for all $k$, because it implicitly assumes some kind of stability of the method. Such results require formal proof.
> >
> > Re W2.2. My comment regarding non-tightness is still valid. I acknowledge that the framework is general, but maybe the analysis can be not tight in some other non-trivial cases, if it cannot recover best-known results in the standard cases.
> >
> > Re W2.3. If I understand correctly, standard (component-wise) clipping still does not fit the updated assumption.

---

### Official Review · Reviewer_GtoB · 2023-11-01

**Soundness:** 3 good
**Presentation:** 3 good
**Contribution:** 3 good
**Rating:** 6
**Confidence:** 3

**Summary:**

The authors analyze a general soft-clipping scheme which extends vanilla clipping schemes such as the one from Eisenman & Stillfjord (2022), and the one from Zhang et al. (2020a). They provide its convergence analysis in the convex and non-convex setting under various assumptions. Furthermore they also compare such clipping schemes, instantiated with several clipping functions, with usual methods for training DNNs, and show that such method can achieve empirical results similar to state of the art methods.

**Strengths:**

### Originality

The results given in the paper are, up to my knowledge, novel. The fact that the method and its proof work for a very general range of clipping schemes makes such proof useful to a larger extend in the literature.

### Quality

Unless I am mistaken, overall the proofs look good and of quality.

### Clarity

The previous works, context, and assumptions for the theorems, as well as the main results (in theory and practice) are clearly described.

### Significance

I believe the problem considered is of interest to the community, since the authors considered the soft clipping scheme method, which has been shown empirically to obtain a good performance in practice, in particular for deep learning.

**Weaknesses:**

- 1. I think that the comparison with state of the arts results (theorems, assumptions), could be made a bit more explicit and structured (see question 1 below)
- 2. I think that the reason why considering new special soft-clipping schemes could be elaborated on further (see question 2 below).

**Questions:**

1. Although the state of the art is well stated which allows one to dig further into the appropriate references, I believe a more explicit *comparison* with the state of the art, gathered at one place in the paper (perhaps in Appendix) would improve the quality of the paper. For instance, I think the two most related papers are the ones from Eisenman & Stillfjord (2022), and the one from Zhang et al. (2020a). Even though by checking those references one may get some idea on the difference between those papers and the submitted one (i.e. Eisenman & Stillfjord (2022) only deal with a strongly convex setting, while Zhang et al. (2020a) have slightly different assumptions, and also, the submitted paper studies a general form of clipping)), I still think it would be good to compare the method exposed in the paper with those two papers more systematically (perhaps in Appendix, in a structured form such as a table) (for instance, saying which algorithms is a special case of which (some algorithms may have momentum, some may have general component-wise functions etc), what assumptions they make in the papers, and what is the resulting convergence rate. For instance, in the strongly convex setting, does the results in the paper retrieve the results from Eisenman & Stillfjord (2022) ? Also, it would be interesting if possible to compare how taking different assumptions and/or considering more general settings actually impacts the proofs compared to the previous ones in the literature.

2. Although a general form of clipping is analyzed, if I am not mistaken, I didn’t see any motivation or related works related to the new gradient clipping schemes from the Appendix A (i.e. Examples 2-4). And in practice it seems that those do not make a big difference (most curves are superimposed so it is hard to differentiate them). Therefore, even though having a general convergence proof is always good, I think it would be even better to further motivate the use of any of the new Examples 2-4 from Appendix A, for instance by mentioning whether they are present in the literature, discussing how their shape depart from the classic soft-clipping (and why it is advantageous), or through experimental results that would show their advantage in some cases.

---

> ### Author Response · Authors · 2023-11-17
> **Response to GtoB**
>
> We thank the referee for the positive remarks and score.
>
> On the Questions:
>
> 1. We have slightly extended the literature review and contributions section, and hope that the differences the referee points out are now more clear. Given how many different settings that could be considered, and how slightly different assumptions lead to significantly different rates, we do not think an exhaustive comparison is feasible. The presented analysis is the first attempt at considering this general class of methods, and we think it is also somewhat unfair to compare this to e.g. highly refined proofs for SGD in special settings. However, in line with the comments by the referee xFTp, we have added some straightforward extensions of the presented analysis on certain fixed step sizes.
>
> 2. As mentioned in the responses to the other referees, we do not yet have a full understanding of which rescalings are best in each situation. Precisely like the referee comments, the error curves are essentially superimposed in both our numerical examples. This indicates that the different rescalings often behave similarly in practice, so that doing some rescaling is more important than what kind of rescaling. The idea of the paper is to simultaneously analyze most such rescalings in a common framework, and finding the optimal choice in a particular situation is still an open problem.

---

> ### Comment · Reviewer_GtoB · 2023-11-22
> **Response to Authors**
>
> Thank you for your answer, which clarifies my questions. I have also read the other reviews and responses, as well as the new revision, and have decided to keep my score: I believe such paper might be of interest to the community, since it analyzes some recent soft-clipping methods in general ways, but I am not an expert in the field, and as such, cannot guarantee full confidence in my evaluation.

---

### Official Review · Reviewer_DnUF · 2023-11-03

**Soundness:** 3 good
**Presentation:** 2 fair
**Contribution:** 2 fair
**Rating:** 5
**Confidence:** 2

**Summary:**

The article introduces a class of soft clipping schemes for various applications which have not been extensively analyzed in the literature, especially in nonlinear cases. It provides a theoretical foundation, demonstrating convergence properties, and highlights that these soft clipping algorithms perform similarly to state-of-the-art methods like Adam and SGD with momentum on large-scale machine learning tasks.

**Strengths:**

1.	The article gives proofs of convergence in expectation with rates in both the convex and the non-convex case.

2.	The numerical experiments in this paper are beautiful which shows that soft-clipping algorithms may offer regularization benefits in cases where other algorithms tend to overfit, encouraging the use of soft-clipping algorithms and further research in the field.

**Weaknesses:**

1.	The comparative analysis with other literatures is insufficient, and it is difficult to see the innovation of the convergence results or proofs in this paper.

2.	This paper lacks some intuitive understanding and analysis of the theorems and corollaries given. Especially for the symbols $\w_k(w)$ without interpretation in Corollary 2, it’s hard for readers to understand and what insight the corollary hopes to provide.

**Questions:**

1.	Have similar convergence analyses been conducted in other literature? If they exist, can you provide a comprehensive comparison to highlight the advantages of the results presented in this paper?

2.	Is it reasonable to assume that $\sum_{k=1}^\infty \alpha_k^2 < \infty$ and $\sum_{k=1}^\infty \alpha_k = \infty$ as stated in Theorem 1? Could you offer a more intuitively understandable explanation for this assumption?

---

> ### Author Response · Authors · 2023-11-17
> **Response to DnUF**
>
> We are happy that the referee likes the numerical experiments and thank them for the fair rating.
>
> On the Weaknesses comments:
>
> 1. We have slightly extended the literature review and contributions section, in line with the comments by referees pggZ and GtoB. If there is some specific additional reference which the referee thinks should also be included, please let us know.
>
> 2. We apologize for the confusing statement of Corollary 2. We did not realize that after some reordering of the material, it was no longer specified that $\omega$ was an element of the sample space $\Omega$. It has now been reformulated. The idea of Corollary 2 is just that the norm of the gradient of $w_k$ will almost surely (i.e. for essentially every random path) go to zero. The min over the $K$ iterations is needed for technical reasons, similar to e.g. the results for standard SGD.
>
>
> On the Questions:
>
> 1. The general class of methods is, to our knowledge new, and there is no convergence existing analysis. With some specific choices of G, we acquire previously analyzed methods like SGD or the TSGD. We have commented on this in the contributions and literature overview sections.
>
> 2. Yes, this is a standard assumption that goes back all the way (at least) to the paper by Robbins and Monro. It means that the step-size sequence goes to zero, but not too quickly. The idea is that we have to take large enough step sizes to actually get closer to the minimum. But once we get close enough, the noise in the stochastic gradient would prevent us from getting closer. By decreasing the step size, the noise is reduced. The assumed step size condition adequately balances these two conflicting demands.

---

### Official Review · Reviewer_pqgZ · 2023-11-08

**Soundness:** 2 fair
**Presentation:** 2 fair
**Contribution:** 2 fair
**Rating:** 3
**Confidence:** 4

**Summary:**

As reminded in the paper, authors such as Mikolov (2013); Duchi et al. (2011); Kingma & Ba (2015) previously proposed element-wise gradient updates. In this paper, the authors recall the soft-clipping approach presented by Zhang et al. (2020a) and propose a class of methods that combine the idea of element-wise gradient updates with soft-clipping. To be more precise, the methods analyzed here are a direct generalization of the “element-wise clipped version” of the soft-clipping algorithm of Zhang et al. (2020a).

Under standard assumptions on the learning rate (Robbins–Monro conditions), standard regularity of the loss function, and standard noise assumptions, the authors prove that the class of stochastic optimizers they proposed converges to a stationary point. In particular, they show that the minimum norm of the gradient converges to 0. All the theoretical results presented in the paper are pretty standard, including the proof-technique aspect: Lemma 2 and Lemma 3, Theorem 1 and Theorem 2 are reminiscent of their respective version for vanilla SGD.

From an experimental aspect, the experiments compare the performance of the stochastic methods introduced in this paper w.r.t. well-known ones such as Adam, SGD + Momentum, and others. The authors conclude that their performances are comparable.

**Strengths:**

**Originality**: The authors present a novel class of stochastic optimizers that combine the idea of soft-clipping and element-wise gradient updates.

**Quality**: The theoretical setting is well-posed and well-presented. Indeed, all assumptions are clearly stated, grounded in the literature, and are not restrictive. Together with their proof, the theoretical results are clearly stated and easy to follow and understand.

**Clarity**: The key messages of the paper are clearly reported at the end of the "Introduction". The readers can follow the whole discussion with little effort.

**Significance**: Given the success of both soft-clipping and element-wise gradient updates, it is important to study if the interaction of these two ideas brings additional non-trivial advantages.

**Weaknesses:**

**Research Aspect:**

While the topic is clearly of interest, I am left wondering about the effective novelty of the contribution. To be more specific, Theorem 3.1 and Theorem 3.2 in Zhang et al. (2020a) already provide convergence results for a hard-clipping algorithm. Additionally, in _Appendix F Soft Clipping_ of the same paper, the authors give a fairly reasonable explanation of why such results should easily generalize to the _soft-clipping_ version of their algorithm.

If I look at Theorem 1 and Theorem 2 of the paper under review, I observe that these are convergence results for optimizers which are a generalization of the “element-wise clipped version” of the soft-clipping algorithm of Zhang et al. (2020a).

While it is clear that these are different algorithms, it is not clear which additional theoretical benefit one inherits by allowing:
1) More general clipping functions (w.r.t. the one in Zhang et al. (2020a));
2) The element-wise clipping itself.

Questions presented later articulate my concerns and provide actionable suggestions on how to improve the contribution of this paper.

**Experimental Validation:**

The experiments presented compare the performance of the stochastic methods introduced in this paper w.r.t. well-known ones such as Adam, SGD + Momentum, and others. Here are some actionable suggestions to improve the experimental aspect of the paper:

1) Figure 1 and Figure 2 lack confidence bars around the lines to represent the uncertainty. They are difficult to read and it would be more interesting to plot those lines in log-scale to clearly see the comparison between the performance of the methods.

2) None of the experiments presented is meant to verify the theoretical insights provided in Section 2. Especially, it would be relevant to verify Theorem 1, Corollary 1, and Theorem 2. There is no need for sophisticated setups: I suggest starting with at least a simple landscape such as a quadratic one: $f(x) = x^{\top} A x$ where $x \in \mathbb{R}^d$ and $A \in \mathbb{R}^{d \times d}$ is or is not an SDP matrix would suffice.
Given that these are the core results of the paper, I would expect even a simple graph where you compare the decay of the minimum of the norm of the gradients w.r.t. the bound that you derive. This would not only validate the theoretical results empirically but also show how loose these bounds are. If anything, at least verifying this for Corollary 1 should be doable as the shape of the bound is very explicit.

3) I suggest adding a comparison of each clipped-component-wise method with its NON-component-wise version: This would help understand the benefit of the new methods and of clipping component-wise w.r.t clipping in a non-component-wise way.

**Quality of the Exposition**:

Regarding style and organization of the text, I would invite the author to reformulate the text written before Section 1 to:

1) Add a section named “Introduction” where you discuss the problem at hand, what you achieved in this paper, and how it is relevant;
2) Add a section called “Related Works” of “Literature Review”;
3) Make sure that the exposition is fluent and pleasant to read rather than a list of points: Separating text in paragraphs is very helpful.

Coming to **Section 1**, currently called Setting, I suggest moving this whole section to the Appendix and just quickly presenting the content of such a section: The assumptions are pretty standard and could be easily summarized in a sentence at the beginning of Section 2 (currently named “Converged Analysis”). For example, one could simply start Section 2 by stating _“In this section, we will provide a convergence analysis for all methods that fit into the setting that is described in the introduction. The assumptions are standard ones from the literature and are reported in Appendix C”_. Of course, if something is NOT standard in the literature, it is worth stating it clearly in the main paper.

Coming to **Section 2**, Lemma 2 and Lemma 3 are not key results and can be moved to the appendix.

Coming to **Section 3**, there is no need to report all the technical details of the numerical experiments in the main paper. I suggest moving the best portion of this section to the appendix.

Coming to the **Appendix**, please consider reporting the statements of the theorems before their proof.

Additionally, there are some typos. Among others, these are the most visible ones:

1) Just below Eq. (3). Please, specify where $(x_i,y_i)$, even if it is $(x_i,y_i) \in \mathcal{X} \times \mathcal{Y}$;
2) On the fifth row from the bottom of page 1: “If the function sufferS”. Add the “s”;
3) At the beginning of page 4: Add the capital letter T to “there” in both lines and add where $x_i$ belongs;
4) Third line from the top of page 3: “stochastic optimization algorithms that mitigateS this issue, is that of performing the gradient update”. Add the “s”.
5) Just above Lemma 2: “but their sharpnessES”. Add the plural. OR, keep it singular and at S after “differ”.

**Conclusion**:

This paper proposes a class of stochastic optimizers based on the combination of soft-clipping and element-wise gradient updates. The authors provide standard convergence bounds under standard assumptions: Their proofs are mostly based on standard techniques applied to the specific case addressed in the paper. The experimental section shows the performance comparison between some of the proposed methods and classic ones such as Adam and SGD + Momentum: The Accuracy and Perplexity are comparable across optimizers. Unfortunately, none of the theoretical results is experimentally validated.
To conclude, the class of optimizers is interesting, but the theoretical analysis provided here is too restricted to be a substantial contribution (questions below might provide inspiration). The experimental side is also lacking and deserves more attention (see above for some suggestions). From an organizational perspective, the manuscript needs significant rewriting (see above for some suggestions).

**Questions:**

Here is a list of questions that I think could guide the authors toward a better final product:

1) What is the behavior of the optimizers w.r.t escaping saddles? What about their preference for sharper or flatter minima?

2) Could you please elaborate on the advantages of component-wise clipping?

2.a) From a **theoretical perspective**,  I am not sure about the actual benefits of component-wise updates. Indeed, you use Assumption 1 only in the proof of Lemma 2 where you show that $\lVert H(\nabla f, \alpha) \rVert < c_h \lVert\nabla f \rVert^2$. Then, you (implicitly) use it also to derive a bound on the expected value of the norm of G. From my perspective, Assumption 1 could be replaced with the following:

i) There exists $c_g \in \mathbb{R}^{+}$ such that $\lVert G(x,\alpha) \rVert < c_g \lVert x \rVert$;

ii) There exists $c_h \in \mathbb{R}^{+}$ such that $\lVert H(x,\alpha) \rVert < c_h \lVert x \rVert^2$.

This way, you could cover more general cases, including yours and also that of Eq. (5) from Zhang et al. (2020a).

2.b) From an **experimental perspective**, in both of your experiments, it seems that Clipped SGD as implemented in Abadi et al. (2015) is the best performer. Is there any case you could find where component-wise clipped methods have better performance than the non-component-wise methods?

What I find missing is a comparison between the optimizer of Eq. (5) (which is without the component-wise clipping) and your component-wise clipped version. In general, for each component-wise clipped method you proposed, I would like to see a clear comparison with its counterpart without component-wise clipping. This would experimentally clarify that having a component-wise clipping is advantageous.

2.c) Of all the component-wise soft-clipping schemes provided, which ones are more advantageous? Is there a way to somewhat understand a recommended shape of the functions $g$ and/or $h$? Which ones clearly show an advantage w.r.t. the one in Eq. (7)? If one were to find one that works better than the one in Eq. (7), how about its NON-component-wise version?

**Minor Questions**:

1) Why is it necessary that $l$ is a non-negative loss function?
2) Can you be more explicit regarding G and H and their mutual relationship?
3) What is the relative size of the constants $B_i$ w.r.t the constant you would find with vanilla SGD?
4) The scheduler used in the experiment is in line with those used in the theoretical results. However, I am sure you noticed that in 150 epochs, it drops from an initial value of $\beta$ to $\sim 0.985 \beta$. It does not strike me as a learning rate that decreases much. How would this compare to a learning rate kept constantly equal to $\beta$?
5) You write that _“We also note that the clipping schemes all require a higher step size for optimal performance than that of both Adam and SGD with momentum, which exemplifies their better stability properties”_. Can you elaborate further on this claim?
6) Could you please make sure that the colors of the lines of different optimizers are consistent between Figure 1 and Figure 2?

---

> ### Author Response · Authors · 2023-11-17
> **Response to pggZ (part 1)**
>
> We thank the referee for the positive remarks and the effort required to write such a comprehensive report.
>
>
> On the Research Aspect comments:
>
> We are aware of the argument in the paper by Zhang et al. (2020a), which has been propagated from a similar argument in previous papers with some of the same authors. As the referee notes, the argument seems "fairly reasonable" when stated in this vague form, but when doing the details it becomes apparent that it is in fact not true. That is, the convergence proof for hard clipping does not translate to a proof of convergence for soft clipping. We have now added a few sentences commenting on this.
>
>
>
> On the Experimental Validation comments:
>
> 1. We unfortunately do not see an easy way to add confidence bars in our computational setup. We also suspect that doing so would further decrease the readability of the figures.
>
> 2. The experiments are intended to illustrate the performance of the methods on real-world applications. The verification of the theorems and corollaries are their proofs in the respective appendices. We aimed for rather complex experiments as this seemed to be what this community wanted to see, but we're happy to include a more simple setup as well. The revised version has two experiments with $\nabla F(w) = Aw$, where we choose $A$ to be a stiff diagonal matrix (large ratio between largest and smallest eigenvalue). The first one provides some motivation for why it is beneficial to use a componentwise method in the stiff case, and the second one compares the different generalized clipping functions. Finally, Figure 1 and 2 are already on a log-scale ("semilogy"), which means that the spacing near 100% and 90% (1 vs. 0.9) is smaller than between e.g. 30% and 40%. We have now changed it to be not logarithmic, which separates the error curves a bit more.
>
> 3. We are not exactly sure what it means to apply non-component-wise versions of the component-wise schemes. Should we replace the component with e.g. the norm of the whole approximate gradient?
>
>
> On the Quality of the Exposition comments:
>
> We have now rearranged the first section along the lines of the referee's suggestion. We are sorry to hear that the referee thinks the text is unpleasant to read, but several of the other referees seem to think that it is easy to follow.
>
> We appreciate the referee's suggestion on different ways to organize the text, but we feel that such drastic changes would lead to a whole new paper. Since the other referees do not argue that the layout is fundamentally broken, we will only make major changes to the first section as mentioned above. One thing we will do is to repeat the theorem statements in the appendices. The mentioned typos will be fixed, except #4. Since the subject of that sentence is "An approach" rather than "optimization algorithms", it should indeed be "mitigates" in the singular rather than plural form.
>
>
>
> On the main Questions:
>
> 1. We think that a proper investigation of this would be interesting, but that it is out of the scope of this paper. However, all of the methods essentially become SGD once the step size becomes small enough, so unless a saddle point is reached very early in the iteration the methods would inherit the properties of SGD.
>
> 2. The new small-scale quadratic example hopefully sheds some light on the benefits of the componentwise setup. Essentially, the idea is that the components of the exact solution will tend to the components of the stationary solution at different speeds, and the componentwise updates allow the approximate solutions to replicate this behaviour.
>
> 2.a) We have now changed Assumption 1 in line with the referee's suggestion.
>
> 2.b) Our opinion is rather that all the methods behave roughly equally well. The minor differences in final loss/accuracy are negligible. We have tried to emphasize this better. The new experiment shows that the componentwise tamed SGD performs better than the non-componentwise version on that particular problem.
>
> 2.c) We do not yet have such an understanding, but we also do not recommend any specific rescaling. The point of the paper is to show that this general class of methods can all be analyzed in a unified way, rather than to suggest a new method that always outperforms other methods. We agree that it would be more satisfying to be able to say that a given rescaling works best for a given problem, but such statements require further research.

---

> > ### Author Response · Authors · 2023-11-17
> > **Response to pggZ (part 2)**
> >
> > On the Minor Questions:
> >
> > 1. It is not; this was a relic from a previous argument that we mistakenly left in. We have removed the word non-negative.
> >
> > 2. We have added a brief comment. G is the chosen rescaling, which completely specifies H.
> >
> > 3. As the other referees pointed out, our error bounds are not tight. With $c_g=1, c_h=0$ we have the SGD method, but the corresponding $B_i$ are larger than what one gets when analyzing SGD directly. The proof technique for the more general case unfortunately does not allow the more specialized algebraic manipulations one could do if $H = 0$.
> >
> > 4. As stated in the paper, $k$ denotes the iteration, not the epoch. We have roughly 469 iterations per epoch, which means that the final step size is rather about 0.125 times the initial step size.
> >
> > 5. With stable we meant in the ODE time-stepping sense, i.e. what time step size can we take such that the approximation stays bounded. The idea is then that we can take larger steps for the clipped methods, meaning that they are more stable. However,  the optimal step size is not the largest possible, and we have now reformulated this statement.
> >
> > 6. Yes, this has been done.

---

> > > ### Comment · Reviewer_pqgZ · 2023-11-22
> > > **Reply to Part 2**
> > >
> > > **Minor Questions**
> > >
> > > 1. Ok, thanks.
> > > 2. Ok, thanks.
> > > 3. I see, thanks.
> > > 4. I am sorry, my bad. Still, it would be interesting to compare the performance with a learning rate kept constantly equal to $\beta$.
> > > 5. Ok, thanks.
> > >
> > > **Conclusion**
> > >
> > > I am still not convinced by the reply provided to me and to the other Reviewers. However, I do appreciate the discussion and I hope the Authors will manage to submit a revised paper that tries to address (some of) the points discussed with me as well as (some of) those discussed with other Reviewers.

---

> > > > ### Author Response · Authors · 2023-11-22
> > > >
> > > > We once again thank the referee for the constructive criticism.
> > > >
> > > > Due to time constraints and other obligations, we will not manage to test the methods on a transformer before the deadline, but we will keep this in mind for the future. We also appreciated the referee's follow-up on the confidence bars, but did not have time to consider this further, since we focused on rewriting the paper to meet the other criticism.
> > > >
> > > > In the new version, we have followed the referee's suggestion to put the assumptions in the supplementary material. We also moved much of the setting description as well as some of the details about the experiment setups there. In addition, as mentioned in the preliminary answer, the first section has been split up into three.
> > > >
> > > > In the new experiment, we compare the non-componentwise standard soft clipping to the componentwise one. We did not test e.g. a non-componentwise arctan scheme, because there was no obvious choice on how to best define such a scheme. We understood what the referee meant with non-componentwise, the question was rather if one should take $\arctan(\alpha \lVert w_k \rVert_2)$ or maybe $\arctan(\alpha \lVert w_k \rVert_1)$ or $\arctan(\alpha \lVert w_k \rVert_{\infty})$. These schemes would likely behave differently, and this is something we would like to investigate more deeply in the future.

---

> > ### Comment · Reviewer_pqgZ · 2023-11-22
> > **Reply to Part 1**
> >
> > I acknowledge and appreciate the reply of the Authors. However, I still have some open points and I need to see the revised version before considering changing my score.
> >
> > **Research Aspect**
> >
> > I am still left wondering what is the theoretical and experimental benefits of these new methods you introduced. Maybe a comparison with the proof for the NON-componentwise clipping could elucidate the theoretical aspect?
> >
> > **Experimental Aspect**
> >
> > 1) While I understand that adding confidence bars might be visually unpleasant, there are two alternatives to go in this direction. The first is to provide a "zoom-in" towards the end of training and add the bars there. The other is to provide a table where you show the average performance measure at the end of training and the average performance measure at test time, and add their standard deviations as well.
> >
> > 2) If you truly wanted to illustrate the performance of the methods on real-world applications, you could have used some little transformers. For example, "https://github.com/karpathy/nanoGPT". There is no need to outperform popular optimizers: If you could show that your algorithms do as well as (or possibly even slightly worse than) the state-of-the-art but with considerably fewer resources (especially less clock time), this would be amazing. I look forward to seeing the new experiment you mentioned in the revised version.
> >
> > 3) Yes, the component-wise clipping is your method for a given $h$ or $g$. The NON-component-wise clipping would be the equivalent method (same $h$ or $g$) where you do not clip component-wise but rather use the same clipping for all the components.
> >
> > **Quality of the Exposition**
> >
> > I truly think that moving all the text that is not key to the comprehension of the main points of the paper should go to the appendix. Repeating standard assumptions, reporting technical lemmas, and presenting the details of the experiments are very likely to distract the reader.
> >
> > Q1: I agree that the intuition you provide makes sense. But, I would not jump to this conclusion so easily: Just like the comment in Zhang et al. (2020a) **sounds** fairly reasonable, the one you presented here **sounds** fairly reasonable as well.
> >
> > Q2: I look forward to seeing the new experiment in the revised version.
> >
> > 2.a: Thanks.
> >
> > 2.b: I look forward to seeing the new experiment in the revised version.
> >
> > 2.c: Thanks.

---

### Author Response · Authors · 2023-11-17

We thank all the referees for valuable feedback. We are in the process of preparing a revised version of the paper and hope to finish this early next week. In the meantime, we have posted individual responses to each referee below which explains how we intend to address the raised points.

---

### Author Response · Authors · 2023-11-22
**Revised version**

We once again thank the referees for the constructive criticism, which we think has improved the paper. We have now uploaded a revised version, where we have tried to meet most of the referee's suggestions, as outlined in the previous preliminary answer.

---

### Meta-Review · Area_Chair_Y2dj · 2023-11-27

**Metareview:**

The paper analyzes a soft version of clipped gradient descent, which was initially discussed by Zhang et al. (2020a), although they did not provide a formal convergence analysis. This ICLR submission focuses on filling this gap by deriving convergence guarantees for the soft-clipping variant of gradient descent.

The initial reviews highlight the incremental nature of the analysis. The paper uses very standard proof techniques and I agree it can be seen as incremental for optimization experts. One could have expected the paper to provide more innovation in a different way, for instance showing benefits from the soft-clipping either theoretically or experimentally. However, this is not the case. From a theoretical point of view, I note that Corollary 1 shows a slower convergence than the standard clipping, see for instance:
Koloskova, Anastasia, Hadrien Hendrikx, and Sebastian U. Stich. "Revisiting Gradient Clipping: Stochastic bias and tight convergence guarantees." arXiv preprint arXiv:2305.01588 (2023).

The reviewers also raised a number of concerns such as lack of motivation for the soft-clipping approach, lack of clarity, lack of some intuitive understanding of the theoretical results, use of restrictive assumptions, etc. The discussion period between the reviewers and the authors did not yield significant changes in the scores assigned to the paper.

Overall, the current version of the paper is below the bar for acceptance. Some of the problems mentioned above could be addressed in a revision. One could use less restrictive assumptions, for instance focusing on (L0, L1)-smoothness which have been shown to benefit clipping methods in https://arxiv.org/abs/1905.11881.

**Justification For Why Not Higher Score:**

The paper is very incremental in nature, this is a standard convergence analysis.

**Justification For Why Not Lower Score:**

N/A

---

### Decision · Program_Chairs · 2024-01-16

Reject